

# Sea spray promotes the sea-to-air transfer of dissolved organic carbon during phytoplankton bloom

Jie Hu[1], Jianlong Li[1], Narcisse Tsona Tchinda[1], Christian George[2,3], Feng Xu[4], Min Hu[4], and Lin Du[1,2,5,*]

[1]Qingdao Key Laboratory for Prevention and Control of Atmospheric Pollution in Coastal Cities, Environment Research Institute, Shandong University, Qingdao 266237, China

[2]School of Environmental Science and Engineering, Shandong University, Qingdao 266237, China

[3]Universite Claude Bernard Lyon 1, CNRS, IRCELYON, UMR 5256, Villeurbanne F-69100, France

[4]State Key Laboratory of Regional Environment and Sustainability, International Joint Research Center for Atmospheric Research (IJRC), College of Environmental Sciences and Engineering, Peking University, Beijing 100871, China

[5]State Key Laboratory of Microbial Technology, Shandong University, Qingdao 266237, China

*Corresponding to*: Lin Du (lindu@sdu.edu.cn)

**Abstract.** The formation of sea spray aerosols (SSA) is linked to wave-breaking events at the sea surface and is widely recognized as an important pathway for the transfer of marine substances to the atmosphere. Although climate change and sea eutrophication have led to the expansion and intensification of coastal phytoplankton blooms, systematic studies on the sea-to-air transfer of dissolved organic carbon (DOC) via SSA during phytoplankton blooms are still lacking, which hinders the understanding of SSA's atmospheric chemistry and climate impacts. In this study, we observed that the phytoplankton bloom can promote DOC enrichment in SSA by 10-fold to 30-fold and investigated the mechanism of DOC sea-to-air transfer using various characterization tools. First, DOC's dynamic accumulation during phytoplankton bloom can significantly impact the interfacial properties of seawater, influencing SSA formation and subsequent DOC transfer. Second, the sea-to-air transfer of DOC depends on its selective enrichment as well as the fractionation process at the air-water interface. Interestingly, the particulate property of operationally defined DOC still needs to be considered during SSA formation. Third, the sea-to-air transfer of DOC is influenced by the synergistic effects of phytoplankton production and heterotrophic microbial processing, rather than being solely dependent on chlorophyll-a concentration. Compared to previous studies, this work focuses on the sea-to-air interface, systematically and comprehensively elucidating the relationships between DOC's transfer mechanisms, biological activity, and SSA formation. This will further improve our understanding of the ocean-atmosphere carbon cycle and provide insights into its impact on global climate change.

## 1 Introduction

As the largest natural reservoir on Earth, oceans serve as a major source of atmospheric aerosols through the release of sea spray aerosol (SSA) (Veron, 2015; De Leeuw et al., 2011). Wave-mediated bubble bursting can produce film drops from the retracting bubble film and jet drops from the ejected water column, and they are major components of submicron SSA (<1 μm) and supermicron SSA (≥1 μm), respectively (Wang et al., 2017; Lhuissier and Villermaux, 2012; Jiang et al., 2022). This



process will release marine substances into the atmosphere. Since SSA can directly or indirectly scatter solar radiation by acting as cloud condensation and ice nuclei, its emission has been suggested to mitigate global warming (Cochran et al., 2017; Ahlm et al., 2017). However, there are still significant uncertainties associated with the effects of SSA on climate, particularly in terms of aerosol-cloud interactions. Although substantial efforts have been made to investigate the formation, composition, and properties of SSA, the complexity of natural environment poses challenges to the comprehensive understanding of SSA.

Dissolved organic carbon (DOC) is typically defined as organic matter that can pass through filters, with pore sizes from 0.2 to 0.7 μm, while the retained fraction is termed particulate organic carbon (POC). DOC comprises approximately 66.2% of the total organic carbon in the ocean and is the dominant organic carbon reservoir in the Earth system (Brooks and Thornton, 2018). As the main engine of the ocean's geo-biochemical cycles, microorganisms and their food webs are the primary sources of marine DOC (Quinn et al., 2015). In sunlit surface seawater, phytoplankton production and heterotrophic microbial consumption of organic carbon are particularly active, leading to higher DOC concentrations and faster turnover. At the same time, SSA formation primarily occurs at the seawater surface, meaning that the organic fraction in SSA is closely tied to biological activity. Previous studies have shown that DOC is typically encapsulated as an organic shell around the sea-salt core (Hu et al., 2024; Song et al., 2024). which not only affects the physicochemical properties of the particles (viscosity, surface tension, reactivity, etc.) (Bertram et al., 2018; Tumminello et al., 2024) but also has a profound impact on their climatic effects (cloud condensation activity, ice nuclei activity, and optical properties) (Xu et al., 2022; Christiansen et al., 2020; Vaishya et al., 2013). However, there are still many uncertainties associated with the sea-to-air transfer pattern of DOC via SSA. On the one hand, marine DOC, as a complex mixture, is estimated to consist of $10^{12}$–$10^{15}$ organic compounds of varying sizes and chemical classes. The large differences in physicochemical properties among these compounds can lead to different transfer patterns. On the other hand, DOC undergoes more active and complex transformation processes in a more productive surface seawater compared to deeper waters, and there is an unclear relationship between DOC composition and biological activity. In key scenarios, such as phytoplankton blooms in coastal areas and climate warming promoting phytoplankton bloom growth (Dai et al., 2023). elucidating the sea-to-air transfer patterns of DOC can enhance predictions of climate impacts and help explore the atmospheric chemistry involving SSA.

In comparison to field observations, laboratory studies offer controlled environments that facilitate the investigation of sea-to-air transfer mechanism of DOC, minimizing the influence of meteorological factors, seawater properties, and external particulate inputs. Previous laboratory simulation studies typically employed simplified modeling systems that focused on a single organic molecule or class of compounds, overlooking the complexity of DOC and the relationship between DOC composition and biological activity. Although a few mesocosm experiments have investigated the effects of phytoplankton activity on the physicochemical properties of SSA (Santander et al., 2023; Jayarathne et al., 2022; Wang et al., 2015), the systematical exploration of the sea-to-air transfer pattern of DOC remains insufficient, hindering our understanding of SSA-cloud interactions. In this study, we conducted experiments with induced phytoplankton blooms in coastal seawater and used



a waterfall type method to simulate the formation of SSA. During the experiments, we monitored DOC and POC concentrations, SSA particle size distributions, and collected samples from different stages of sea-to-air transfer, including bulk seawater, sea surface microlayer, submicron SSA, and supermicron SSA. DOC in the different samples was characterized using multiple techniques, and the sea-to-air transfer pattern of DOC via SSA during the phytoplankton bloom was systematically investigated.

## 2 Experimental sections

**2.1 Phytoplankton Bloom**

Seawater was collected on May 31, 2024, at Shazikou Pier (120°33'28"E, 36°6'37"N) Qingdao, China, and immediately transported to the laboratory. There, it was filtered through a 1-mm mesh sieve and transferred into 30 clear polycarbonate containers, each with a capacity of 28 liters. A 4-fold diluted Guillard's F medium was added to each container, and these containers were placed outdoors on a flat to promote phytoplankton blooms under natural sunlight (Fig. S1). The phytoplankton

bloom experiment began on June $1^{st}$, 2024, lasting for 18 days (the outdoor temperature was $22 \pm 3$ ℃). During this period, 10 simulation experiments on nascent SSA were conducted.

**2.2 Generation and Collection of Nascent SSA**

In each experiment, three containers of seawater (84 liters) were filtered by 50 μm mesh screen to remove large particles and phytoplankton aggregates and were introduced in our home-made SSA simulation tank (length × width × height = 0.6 ×

0.5 × 0.6 $m^3$). The SSA simulation tank has a design similar to that of the Marine Aerosol Reference Tank, which produces nascent SSA through the plunging waterfall (Stokes et al., 2013). More parameter comparisons are provided in Table S1. Although the intermittent plunging waterfall mode was shown to better reproduce SSA generation, we used a continuous plunging waterfall in order to improve the sampling efficiency of SSA. To eliminate the influence of seawater temperature, all SSA generation experiments were conducted at 25℃. More details on SSA generation are provided in the Supplement. Nascent

SSA was transported with purified air (Zero Air Supply, Model 111, Thermo Scientific), and the airflow was dried to a relative humidity below 30% (Monotube Dryer, MD700-12F-3, Perma Pure, USA) before collection and measurement. Samples of single particles were collected by a single particle sampler (DKL-2, Genstar electronic technology Co., Ltd., China) and then analyzed by transmission electron microscopy (TEM, FEI Tecnai G2 F20, Thermo Fisher Scientific, USA). Using a low-pressure cascade impactor (DLPI+, Dekati Ltd., Finland), nascent SSA particles were collected with 14 different particle size

classifications (Table S2) and distributed into submicron SSA (0.016-0.94 μm) and supermicron SSA (1.62-10 μm) samples. These samples were extracted with ultrapure water (>18.2 MΩ·cm, 25 ℃, Millipore) and filtered with 0.45 μm filters. Further collection details are provided in the Supplement. Blanks were prepared by unexposed quartz fiber filters with the same treatment as for SSA samples.

Seawater was collected at a depth of 10 cm in each container and immediately filtered at low pressure (≤0.2 MPa, avoiding



the Chl-a loss) through a GF/F filter (47 mm, Whatman, UK). Both filters and filtered seawater were stored at -20 °C in a dark

environment. Sea surface microlayer (SML) was collected in the SSA simulation tank using the glass plate method (Hu et al.,

2024). SML samples were filtered through a 0.45 μm filter and then stored in a dark environment at -20 °C. Ultrapure water

was treated in the same way as procedural blanks for seawater and SML samples.

### 2.3 Aerosol Characterization and Chemical Analysis

**2.3.1 SSA Particle Size Distribution**

Particle size distributions of dried SSA were measured by a scanning mobility particle sizer (SMPS, GRIMM, Germany)

and aerodynamic particle sizer (APS 3321, TSI, USA). SMPS was operated at a sampling flow rate of 0.3 L min$^{-1}$ and a scan

rate of 5 min, providing the particle size distribution with electrical mobility diameter ($d_{em}$) between 0.02 and 1 μm. APS

detects SSA particles with aerodynamic diameters ($d_a$) ranging from 0.5 to 10 μm, and the scanning period is set to 1 min.

Assuming that SSA particles are spherical, their physical diameter ($d_p$) is related to $d_e$ and $d_a$ by the following (Eq. (1)) (Harb

and Foroutan, 2022; Stokes et al., 2016):

$$d_p = d_{em} = \frac{d_a}{\sqrt{\frac{\rho_{eff}}{\rho_0}}} \qquad (1)$$

where $\rho_0$ is referred to unit density (1.0 g cm$^{-3}$), and $\rho_{eff}$ is the effective density of the particles (2.0 g cm$^{-3}$), determined

experimentally. Two particle size distributions from SMPS and APS were integrated at about 1 μm.

**2.3.2 Chl-a, POC, DOC, Sodium Ion and Surface Tension of Seawater**

The concentration of POC in seawater was determined using an elemental analyzer (Elementar, UNICUBE), which

measured the POC content in a 1 cm diameter circular area on the GF/F filter. Chl-a in the remaining filters was extracted with

90% (v/v) acetone for 24 h at 4 °C in the dark. Fluorescence values were measured using a Turner Designs 10AU Field

Fluorometer (USA), calibrated with chlorophyll standards (Sigma-Aldrich Co.) (Zhong and Ran, 2024; Rocchi et al., 2024),

and the values were thereafter converted to Chl-a concentrations in the corresponding volume of seawater. The concentration

of DOC in the sample was measured by high temperature catalytic oxidation using a total organic carbon (TOC) analyzer

(TOC-5000, Metash, China). Measurements were repeated at least 3 times with a relative standard deviation of less than 3%.

Sodium ion (Na$^+$) in the samples were measured using an ion chromatograph (Dionex ICS-600, Thermo Fisher Scientific,

USA). The surface tension of filtered seawater and SML samples was measured by the platinum plate method using a surface

tension meter (POWEREACH, JB99B, China).

### 2.3.3 Measurements of DOC's Fluorescence

The excitation-emission matrix (EEM) of DOC was obtained using a fluorescence and absorbance spectrometer (Duetta™,

Horiba Scientific, Japan). The excitation wavelength of EEM was in the range of 250-620 nm, the emission wavelength was



in the range of 250-700 nm, the scanning intervals were set to 5 nm and 2 nm, respectively, and the slit width was fixed at 5

nm. Ultrapure water was used as a blank reference, and all EEM results were normalized to Raman units (R. U.) by the Raman

peak of water (Ex=350 nm) (Chen et al., 2023). EEM data analysis using parallel factor analysis (PARAFAC) with non-

negativity constraints were performed with the DOMFlour toolbox by MATLAB R2020a (Stedmon and Bro, 2008).

### 2.3.4 Measurements of Saccharides

Saccharides with molecular weight lower than 1 kDa are typically monomers or oligomers consisting of less than five

monomers. Due to the high biological turnover, their concentrations represent only 1-2% of DOC in seawater(Kaiser and

Benner, 2009), and they are almost undetectable in SSA(Jayarathne et al., 2016). Thus, we focused on saccharides with

molecular weight higher than 1 kDa in SSA and seawater samples, as these are key components of transparent exopolymer

particles, and they constitute most of algae-derived high-molecular-weight (>1 kDa) DOC. The samples were subjected to

dialysis for desalting, followed by acid hydrolysis, nitrogen blowing, and re-solubilization (Engel and Händel, 2011).

Saccharides were hydrolyzed to monosaccharides, followed by detection using high-performance anion exchange-

chromatography with pulsed amperometric detection (HPAEC-PAD, ICS 6000, Dionex), coupled with a Dionex CarboPac

PA20 column (2×250 mm) and a Dionex CarboPac PA20 guard column (2×50 mm). NaOH and sodium acetate (NaAc) were

used as mobile phases at a flow rate of 0.250 mL min$^{-1}$. The detailed gradient elution procedure is shown in Table S3.

Identification of the saccharides was based on the retention times of 16 standards (glucose, fructose, xylose, galactose, mannose,

trehalose, fucose, rhamnose, arabinose, xylitol, arabinitol, mannitol, galactosamine, glucosamine, galacturonic acid and

glucuronic acid). The quantification was performed using seven-point standardized calibration curves with concentrations

ranging from 10 nM to 10 μM.

### 2.3.5 Characterization of DOC using Ultra-High Resolution Mass Spectrometry

Three sets of samples from the early, peak, and end phases of the phytoplankton bloom were pre-treated for desalting and

concentrating using a PPL solid-phase extraction (SPE) column (100 mg/3 mL, Agilent Technologies). The treated samples

were then separated for organic compounds using gradient elution (see Table S4) on an ultra-high-performance liquid

chromatography (UHPLC) system (UltiMate 3000, Thermo Scientific), which was coupled to an LTQ-Orbitrap Velos Pro ETD

(Thermo Scientific) operating in ESI- mode to obtain m/z signals from 150 to 1000. Blank samples were processed using the

same procedure. The m/z signals from the corresponding blank samples were subtracted from the aerosol and seawater samples

using Xcalibur 4.2.1 software, and the remaining m/z signals were assigned molecular formulas by MFAssignR (R version

4.3.2)(Schum et al., 2020; Radoman et al., 2022). Further details on sample pretreatment, instrumental conditions, and

assignment principles are given in the Supplement.





## 3 Results and discussion

### 3.1 Variations of Chl-a and DOC during the Phytoplankton Bloom

As an indicator for phytoplankton growth, the time series of Chl-a concentration revealed that a phytoplankton bloom occurred during the experiment (Fig. 1a). The Chl-a concentration increased from 0.48 μg L$^{-1}$ to 36.02 μg L$^{-1}$ on day 9, and then decreased to a minimum of 6.53 μg L$^{-1}$. However, the DOC concentration in seawater did not follow the same trend as that of Chl-a, decreasing from $674.9 \pm 64.6$ μM to $307.39 \pm 66.58$ μM on day 5, before increasing again (Fig. 1b). The initial decline in DOC concentrations was also observed in each of the other six experiments, and these declines always ceased after

the depletion of inorganic nitrogen and phosphorus (Biermann et al., 2014). It has been reported that the addition of inorganic nutrient not only promotes phytoplankton blooms, but also enhances bacterial production and respiration rates, which increases their ability to utilize DOC(Carlson et al., 2004; Jiao et al., 2010; Cai and Jiao, 2008). Given the slow increase in Chl-a concentration and the rapid rise in POC concentration over the first five days (Fig. 1c), the initial decline in DOC concentration may result from depletion by heterotrophic microbial growth. The high bacterial activity in seawater on day 5 was further

confirmed by subsequent results characterizing bacterial activity through saccharide concentrations (this will be further discussed in section 3.6). In our experiment, DOC constitutes the major component of the total organic carbon pool (TOC = POC + DOC). Variations in DOC concentration during the phytoplankton bloom have the potential to affect the sea-to-air transfer of DOC via SSA.

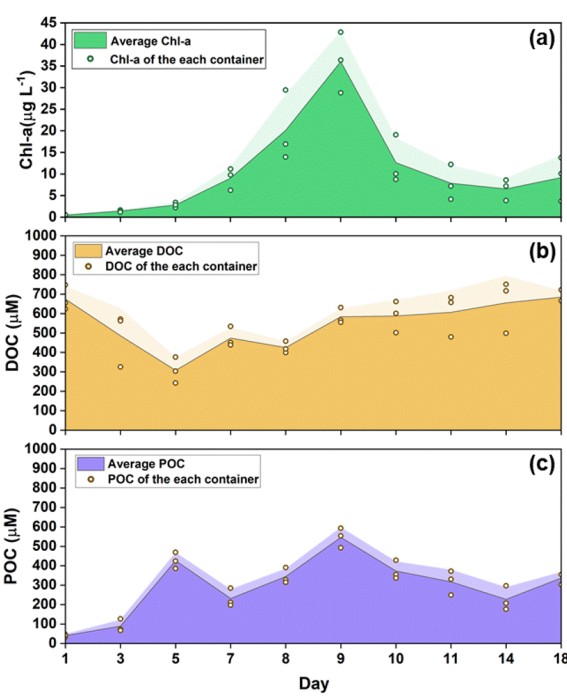

**Figure 1.** Time series of physicochemical properties of seawater during the phytoplankton bloom. (a) chlorophyll-a (Chl-a),



(b) dissolved organic carbon (DOC), and (c) particulate organic carbon (POC) concentrations in seawater. Scattered points represent measurements of seawater from each container. Mean and standard deviations are for three containers of seawater in each experiment.

### 3.2 Effects of DOC Variations on SSA Formation

The distributions of SSA particle size during the phytoplankton bloom are shown in Fig. 2a-b. Before the Chl-a peak, the production of submicron SSA decreased with increasing Chl-a concentration; after the Chl-a peak, it exhibited a slight increase as Chl-a concentration decreased. Supermicron SSA showed an opposite trend. The trend of SSA number concentration closely followed that of submicron SSA, as submicron SSA mainly contributes to the number concentration. In general, the phytoplankton bloom reduced the number concentration of SSA. The geometric mean diameter of SSA increased from 103.8

± 5.0 nm to a peak of 136.3 ± 5.4 nm, before decreasing to 115.0 ± 6.9 nm (Fig. 2c). The dynamic accumulation of DOC during phytoplankton blooms will have a significant impact on bubble bursting and SSA formation by modifying seawater properties.

As an important surface property, surface tension has been proven to be an influential parameter in controlling bubble bursting and SSA formation (Tammaro et al., 2021; Sellegri et al., 2006). Theoretically, the surface tension of seawater is closely related to the composition and physicochemical properties of the SML. Therefore, the surface tension was measured

for the SML samples, with bulk seawater samples taken as controls (secondary formation of the SML within the samples was not considered). As shown in Fig. 2d, the surface tension of SML increased rapidly from day 1 to day 5, then stabilized. The work on the Langmuir monolayer showed that the presence of organic matter reduces the surface tension by increasing the average molecular area and weakening hydrogen bonding between water molecules at the air-water interface (Xu et al., 2023). However, the results shown in Fig. 3 demonstrate that the increase in surface tension in the SML samples is due to the

accumulation of DOC during the phytoplankton bloom. Several studies have found that phytoplankton blooms can result in the formation of mucus on the water surface, which is typically an excessive accumulation of extracellular polysaccharides (Ternon et al., 2024; Medina-Pérez et al., 2021). In contract, this can increase the viscosity of SML and potentially enhance its surface tension (Jenkinson and Sun, 2010). From day 1 to day 5, the rapid increase in the surface tension of SML samples appears to be related to the rise in their saccharide concentration (see in Fig. 6a).

The surface tension of SML exhibited a significant correlation with the number concentration ($r = -0.881$, $p < 0.01$) and the geometric mean diameter ($r = 0.929$, $p < 0.01$) of SSA. Increases in surface tension and viscosity are reported to inhibit the instability of bubble film edge and the development capillary waves during bubble bursting, thereby reducing the number of film drops produced by bubbles and increasing the droplet size (Wang and Liu, 2025; Lhuissier and Villermaux, 2012). High surface tension strengthens the attraction between liquid molecules at the surface, making the bubble film more able to

withstand the pressure difference between the inside and outside of the bubble (Wu et al., 2021; Wu et al., 2022). This resulted in an extended bursting time of the bubbles population, leading to the appearance of a long-lasting foam layer on the water

surface from day 8 to day 18 (Fig. S2), which inhibited the production of SSA. Additionally, statistical analysis of jet drops

during bubble bursting demonstrated that surface tension and viscosity worked synergistically to control both the number and

size of the droplets produced (Berny et al., 2021). Increased surface tension and viscosity in this experiment appear to promote

the production of jet drops. Overall, phytoplankton bloom significantly impacts SSA formation by altering the physical

properties of the seawater interface.

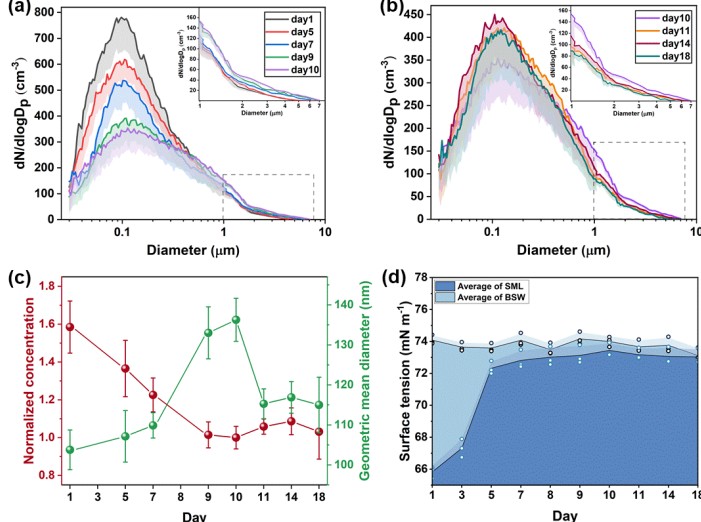

**Figure 2.** Time series of SSA formation during the phytoplankton bloom. (a) Particle size distributions of SSA from D1 to

D10; (b) Particle size distributions of SSA from D10 to D18. In order to show clearer results, the gray dashed box area is the

result of the aerodynamic particle sizer and is enlarged to the upper right corner. Shading is the standard deviation in the

negative direction. (c) Number concentrations and geometric mean diameters of SSA. Error bars are standard deviations for

all measured values. (d) Surface tension of SML and bulk seawater. Shading is the standard deviation of three repeated

measurements.

### 3.3 The Phytoplankton Bloom Promotes DOC Enrichment

The time series of DOC concentration in the SML (Fig. 3) exhibits a slightly positive correlation with that of POC

concentration in seawater ($r =0.761$, $p=0.054$). This suggests that DOC in the SML during the phytoplankton bloom may share

a similar origin with POC in seawater or may result from the initial degradation of POC (Crocker et al., 2022; Jiao et al., 2010).

It is assumed that the concentration of $Na^+$ is constant during the sea-to-air transfer. The enrichment factor (EF) can quantify

the degree of organic matter enrichment in this transfer. It is defined as the concentration ratio of the target substance (X) to

that of $Na^+$ in SSA particles or SML relative to the ratio in seawater (Eq. (2)):

$$EF = \frac{(X)_{SSA\ or\ SML}/(Na^+)_{SSA\ or\ SML}}{(X)_{SW}/(Na^+)_{SW}} \qquad (2)$$





Due to the limited SSA collection, samples on day 1, day 9, and day 18 were not analyzed for the EF of DOC and were

only used for subsequent mass spectrometry analysis. As shown in Fig. 3, the highest EF in the SML was observed on day 5,

while that in SSA was on day 7 (given the low EF of DOC in the SML on day 9, it is unlikely that DOC's EF in SSA at this

time would reach its highest value). The EFs of DOC in SML, supermicron SSA, and submicron SSA can increase by up to

~4-fold, 10-fold, and 30-fold, respectively, during the phytoplankton bloom. Meanwhile, the morphological structural images

of SSA illustrate a significant enhancement in DOC enrichment (Fig. S3). However, the time series of DOC's EF in the SML

and SSA do not match that of Chl-a concentration, indicating that the sea-to-air transfer of DOC is not solely dependent on

phytoplankton abundance. Temporal fluctuations in DOC composition and structure triggered by biological cycles during

phytoplankton blooms may play an important role in influencing DOC's sea-to-air transfer.

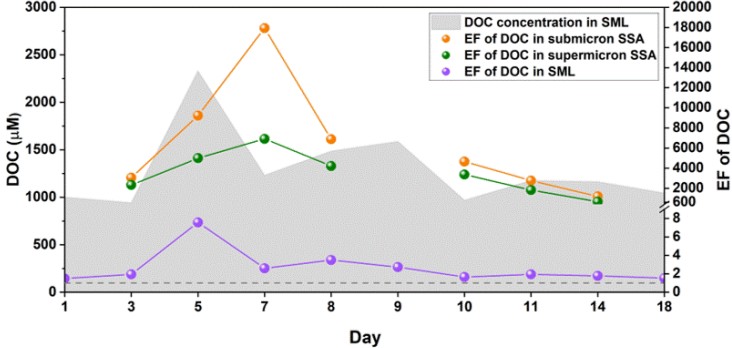

**Figure 3.** Time series of DOC enrichment during the phytoplankton bloom. Enrichment factors of DOC relative to $Na^+$ in the
SML (purple), submicron SSA (orange) and supermicron SSA (green). The gray background is the concentration of DOC in
the SML.

**3.4 Overview of Organic Molecules during the Sea-to-air Transfer**

To investigate the link between the sea-to-air transfer of DOC and biological activity, samples from the early (day 1),

peak (day 9), and late (day 18) stages of the phytoplankton bloom were selected for mass spectrometry analysis. Molecular

formula assignments of m/z lists from different samples were carried out using MFAssignR. The molecules were categorized

into CHO, CHNO, CHOS, and CHNOS groups and plotted in van Krevelen diagrams (Fig. S4). Among the different samples,

the most prevalent molecular formulas were from the CHO and CHNO groups, which accounted for (23.19 ± 3.34) % and

(63.49 ± 4.52) % of the total number of assigned molecular formulas, and (27.94 ± 7.53) % and (65.17 ± 8.18) % of the total

intensity of assigned molecular formulas, respectively (Fig. 4a). Compared to day 1 and day 18, the differences in the number

and intensity of assigned molecular formulas between SML and SW samples are greater on day 9. DOC molecules exhibited

a higher intensity-weighted average value of H/C and a lower value for O/C on day 9 (Table S5), which indicates that more

low-oxidized and hydrophobic organic matter has been produced, making it more readily enriched in the SML. According to

Fig. 1a above, despite the significant decline in submicron SSA production on day 9, 2201 molecular formulas were assigned,



much higher than 948 molecular formulas assigned to supermicron SSA whose production was near its peak at the same time. This reflects the fact that DOC produced during the bloom peak is more easily transferred into submicron SSA. Although the intensity-weighted average H/C value for submicron SSA did not show a significant difference compared to supermicron SSA

(Table S5), using aerosol mass spectrometry coupled with positive matrix decomposition, Wang et al. found that organic matter in submicron SSA did have higher H/C values (Wang et al., 2015).

Figure 4b presents Venn diagrams that illustrate the number of assigned molecular formulas in the four sample types, with the intersections indicating shared molecular formulas. By comparing the samples two by two, submicron SSA and the SML showed greater molecular similarity $(33.2 \pm 7.8)$ %, and the highest percentage of identical molecular formulas was observed

at 41.7% on day 9. The above results reflect that the SML, as a crucial region for air-water interface fractionation of DOC, is highly sensitive to variations in DOC composition in seawater and serves as an important source of DOC in submicron SSA. As shown in Fig. 4a, from the three stages of phytoplankton bloom, the total intensity of assigned molecular formulas in seawater remained relatively stable, while the total number showed a stepwise decrease. The reduced molecular diversity is likely to further affect the sea-to-air transfer of DOC. For instance, the proportion of shared organic molecular formulas in SW,

SML, submicron SSA, and supermicron SSA increased from 12.4% on day 1 to 16.2% on day 9 and 26.3% on day 18.

Based on the van Krevelen diagrams (Fig. S4) and a more detailed division rule (Fig. S5), the assigned molecular formulas in the different samples and their groups (CHO, CHNO, CHOS, and CHNOS) can be further allocated into seven biochemical categories (Suo et al., 2024; He et al., 2023). As illustrated in Fig. 4c, lipid-like, protein-like, carbohydrate-like and lignin-like molecules accounted for the majority of both the relative abundance and the number of molecular formulas. Previous studies

have confirmed that the DOC produced by algae consists of two major aliphatic groups: proteins and polysaccharides (Suo et al., 2024). Lignin-like molecules are mainly derived from terrestrial vascular plants (Osburn and Stedmon, 2011) and metabolites of precursors secreted algae (Labeeuw et al., 2015). which has been widely considered to predominantly contribute to humic-like substances (HULIS) (Kim et al., 2003). However, their relative abundance does not directly reflect the true variation in their concentration, considering the effects of ionization mode and ionization efficiency. Therefore, additional

methods were taken to quantify the concentration fluctuations of protein, saccharides and HULIS in DOC to better understand the link between DOC's sea-to-air transfer and biological activity.



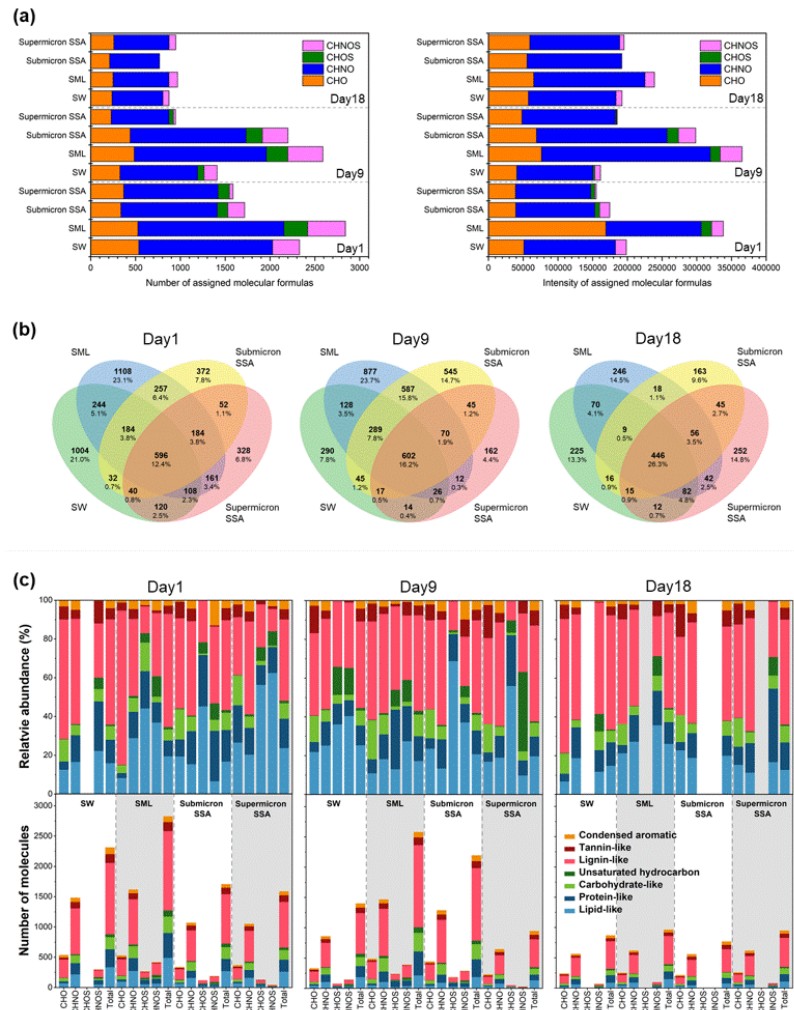

**Figure 4.** Sea-to-air transfer of organic molecules. (a) The number and intensity of molecular formulas assigned in different samples. (b) Venn diagram illustrating the number of assigned molecular formulas. Percentages represent the proportion of molecules in each region relative to the union set. (c) Relative abundance and molecular number contributions of the seven molecular types derived from the Van Krevelen diagrams. The gray background is for visual differentiation purposes only.

## 3.5 Selective Enrichment and Interfacial Fractionation of DOC

Three fluorescence compounds co-exiting in SW, SML, submicron SSA, and supermicron SSA were identified by split-half validation using the EEM-PARAFAC method (Fig. 5a). The peaks of protein-like substances (PRLIS) are mainly at (280 nm)/ (330 nm), and most of them were due to tryptophan-like substances (Santander et al., 2022). The HULIS 1 peaks mainly appear at the excitation/emission wavelengths of (<245 nm or 320 nm)/ (396 nm), and the HULIS 2 peaks mainly appear at (260 or 360 nm)/ (450-455 nm). It has been reported that HULIS 2 is a photooxidation product and, therefore, has a higher



oxygen content than HULIS 1 (Santander et al., 2023; Barsotti et al., 2016).

The EEM intensities of the three fluorescence compounds in each sample type fluctuated during the phytoplankton bloom (Fig. 5b). To exclude the correlation between EEM intensities in SSA samples and the mass of collected SSA, we standardized the EEM intensities using $Na^+$ concentration. During phytoplankton blooms, HULIS1 and HULIS2 gradually accumulated in seawater, while PRLIS initially increased and then decreased. The EEM intensity of PRLIS was significantly stronger than that of HULIS1 and HULIS2 in the SML, submicron SSA, and supermicron SSA. PRLIS includes small peptide molecules and soluble amino acids formed from the degradation of cells, cellular debris, or large proteins. They contain both hydrophilic groups ($-NH_2$ and $-COOH$) and hydrophobic carbon chains, thus having strong enrichment potential during the fractionation at the air-water interface. This high enrichment capacity aligns with the previous findings (Triesch et al., 2021b; Triesch et al., 2021a). The time series of PRLIS's intensity matched with that of DOC's EF in the SML and SSA (Fig. 3), indicating that PRLIS was likely the primary contributor to the increase of DOC's EF during the phytoplankton bloom. Moreover, as a less oxidized organic matter, HULIS1 exhibited a greater enrichment capacity in SSA than HULIS2. Consequently, PRLIS and HULIS1 have greater abundance in the SML and SSA compared to HULIS2 (Fig. 5c). Compared to supermicron SSA, the EEM intensities of the three organic compounds in submicron SSA is higher. Besides the properties of the organic matter itself, the sea-to-air transfer of DOC is also influenced by the generation mechanism of SSA. Before the bubble film ruptures at the water surface, gravity continuously expels the liquid within it, while surface-active substances, being lighter, are pushed upward, forming a vanishingly thin film (Lhuissier and Villermaux, 2012). The resulting film drops are thus enriched with a higher concentration of organic matter. In contrast, jet drops primarily originate from the liquid at the air-water interface inside the bubble and are typically less enriched in organic matter than film drops.

The correlations between organic carbon concentration and the three fluorescent compounds during sea-to-air transfer were further explored (Fig. 5d). PRLIS, HULIS1, and HULIS2 maintained significant or near-significant positive correlations within the same sample of sea-to-air transport; however, their correlation weakened across different samples. In seawater and SML samples, the weakened correlation may be due to the fact that these three organic fractions originate from different organic carbon pools in seawater. We found that the EEM intensities of the three compounds in seawater are positively correlated with the DOC concentration, while in the SML, these compounds are significantly positively correlated with the POC concentration in seawater. This further supports the idea that the DOC in the SML (as discussed in section 3.3) likely originates from POC in seawater rather than DOC. For submicron and supermicron SSA, the weakening of the correlation may result from DOC undergoing different air-water interfacial fractionation processes. For instance, the sea-to-air transfer of DOC via submicron SSA is likely linked to multiple air-water interfaces, such as underwater bubbles, SML, and bubble films, where the selective enrichment of organic compounds results in weak or even negative correlations with those in the other samples. In contrast, DOC in supermicron SSA undergoes weaker air-water interfacial fractionation processes compared to submicron SSA. The results in Fig. 5d show that the correlation between the EEM intensities of these compounds in submicron SSA and



those in the other samples is the weakest. However, in supermicron SSA, they still exhibit good positive correlations with

those in seawater. Another factor that may reduce the correlation among the samples is the time lag in the sea-to-air transfer.

For instance, PRLIS peaked on day 5 in the SML, while it peaked on day 7 in submicron SSA. This lag could be related to

bacterial activity or may be limited by the size of the DOC in the SML.

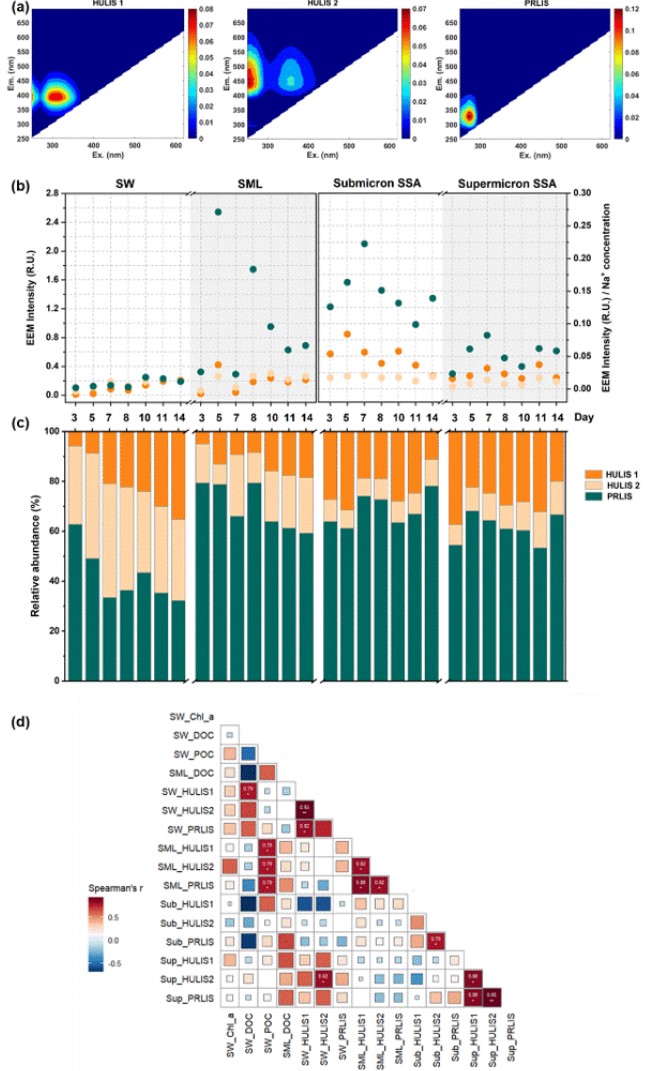

**Figure 5.** Sea-to-air transfer of HULIS and PRLIS. Three organics identified using the EEM-PARAFAC method: (a) HULIS

1, HULIS 2, and PRLIS. (b) EEM intensities of the three organics in different samples with respect to time. Note that in order

to exclude the effect of SSA collection mass on EEM intensity, EEM intensities of SSA samples were normalized with their

$Na^+$ concentrations. (c) Relative abundance of EEM intensities of the three organics in different samples with respect to time.

(d) Spearman's correlation between Chl-a, DOC and POC concentrations in seawater, POC concentration in the SML and EEM

intensities of three fluorescent substances.



### 3.6 Bacterial Modification and Size Limitation of DOC

Phytoplankton typically sequesters excess carbon as saccharides in energy storage materials, cell walls, and extracellular polysaccharides, and this process is influenced by phytoplankton growth, heterotrophic bacteria, and environmental conditions. The accumulation of saccharides in seawater and phytoplankton growth processes are interrelated but not fully synchronized.

In both seawater and SML, two distinct peaks in total saccharides were observed, occurring before and after the peak of Chl-a concentration (Fig. 6a). It has been reported that under inorganic nutrient-sufficient conditions, the high metabolic activity of phytoplankton during the exponential growth phase leads to the secretion of large amounts of extracellular polysaccharides, which rapidly aggregate into highly viscous, transparent exopolymer particles (TEP), thereby promoting phytoplankton clustering (Passow, 2002; Villacorte et al., 2015). The direct conversion of DOC into TEP by bacteria during the early stage of

phytoplankton blooms also represents an important pathway for TEP production (Engel et al., 2004; Passow, 2002). Since added inorganic nutrients were depleted on day 5, phytoplankton is likely to adjust its metabolism, leading to reduced extracellular polysaccharide secretion, while enhancing bacterial degradation of saccharides (Passow, 2002). A second peak in saccharide concentration occurring on day 11, is attributed to the collapse of the phytoplankton bloom, which releases intracellular saccharides and causes a temporary rise in its concentration in seawater (Mühlenbruch et al., 2018). Later, bacterial

degradation of these saccharides leads to a decrease on day 14 (Hasenecz et al., 2020).

As shown in Fig. 6b, the total saccharide concentration accounted for $32.84 \pm 10.02$ % of DOC in seawater and $28.23 \pm 18.00$ % of DOC in the SML. The bacterial activity was indicated by the ratio of the sum of fucose and rhamnose concentrations to the sum of arabinose and xylose concentrations, with ratios below 1 indicating high bacterial concentrations (see the black scattered points in Fig. 6b) (Jayarathne et al., 2022). Seawater and SML exhibit different time series of bacterial activity, suggesting that they may harbor distinct microbial communities. The rapid increase in the percentage of saccharides in the

SML after day 10 was accompanied by an increase in bacterial activity within the SML. This increase in saccharide percentage can be attributed, on one hand, to the release of saccharides from phytoplankton die-off, and on the other hand, to a decrease in PRLIS concentration in the SML (Fig. 5b), which reduced the competition for saccharides to be enriched at the interface. However, the percentage of saccharides in SSA remained consistently below 10% and did not show a significant increase until

day 14, corresponding to the peak bacterial activity in the SML. Hasenecz et al. found that further addition of heterotrophic bacteria significantly increased saccharide enrichment in SSA, as the enzymes released by these bacteria further modified the saccharides (Hasenecz et al., 2020). The time lag in the increase of saccharide percentage from seawater to SSA through SML indicates that bacterial modifications are crucial in the sea-to-air transfer of saccharides.

Saccharides are more abundant in seawater and the SML, with nearly all of the 16 monosaccharides/disaccharides

detectable in the hydrolyzed products of polysaccharides (Fig. 6c). Due to the interconnectivity between seawater and SML, their saccharide compositions show good similarities. However, the varying fractionation behaviors of saccharides during the



sea-to-air transfer led to a reduction in their diversity in SSA. Glucose, mannose, and xylose emerged as the most relatively abundant saccharides in SSA, with corresponding EFs being the highest in both submicron and supermicron SSA (Fig. S6). In contrast to some simplified experimental model systems (Xu et al., 2024; Hasenecz et al., 2019). the EFs of saccharides in submicron and supermicron SSA do not show the expected differences in our experiments (Fig. S6). Furthermore, a substantial variability exists in the EFs among different saccharides. We suggest that the size limitations of saccharides should be considered during the sea-to-air transfer via SSA, despite their operational definition as DOC (<0.45 μm).

According to the bubble-mediated mechanism of SSA formation, the size of polysaccharides may play a key role in their enrichment in SSA. Regardless of whether the film drops are generated by Rayleigh-Taylor (Lhuissier and Villermaux, 2012) or Squire instability (Jiang et al., 2022), we believe that the main prerequisite for polysaccharides to enter the film drops is their consistent presence in the bubble film, at least until it ruptures. For bubbles with radius of ~1 mm, the bubble film thickness can reduce to around ~100 nm at rupture (Lhuissier and Villermaux, 2012). However, bubbles with radius smaller than ~1 mm dominate the number concentration (Stokes et al., 2013), and their bubble film will be thinner at rupture. A similar theory applies to the generation of jet drops: the "liquid layer" (the thickness of which is determined by viscous, inertial, and surface tension forces) (Ji et al., 2022) located at the air-water interface inside the bubble needs to hold the polysaccharide until the cavity collapses to form jet drop.

Generally, polysaccharides that can be enriched in film drops are smaller in size compared to those enriched in jet drops, due to the thinner bubble film and smaller film drop. The size of saccharides is partially determined by their functionality. Glucose and fructose are important monosaccharides that constitute cellular energy storage substances (glucan and fructan), and it is reported that the majority of these polysaccharides have sizes smaller than 6 nm in phytoplankton blooms (Hasenecz et al., 2020). Therefore, they exhibit higher EFs in both submicron and supermicron SSA (Fig. S6). For a specific type of polysaccharide, if its size does not meet the conditions for entry into film drops but does for jet drops, it is reasonable to observe its greater EF in jet drops than in film drops. For example, fucose is nearly undetectable in submicron SSA samples and is only enriched in the SML and supermicron SSA. During phytoplankton blooms, fucose-constituted polysaccharides primarily range from 50 nm (≈100 kDa) to 450 nm (Jayarathne et al., 2022) and are resistant to bacterial hydrolysis (Murray et al., 2007). making them difficult to be enriched in submicron SSA. Saccharide alcohols, including xylitol, arabinitol, and mannitol, make up less than 5% of the total saccharides. These compounds are effective osmoregulatory substances that help phytoplankton cells adapt to their environment. They may also result from bacterial decomposition of organic matter, typically in the free state or as low molecular weight forms (Jayarathne et al., 2022; Pramanik et al., 2011). and can be enriched in SSA. Glucuronic and galacturonic acids are the main components of bacterial extracellular polysaccharides but they make up only a small percentage of those of phytoplankton (Zhang et al., 2015; Bhaskar and Bhosle, 2005). Since the polysaccharides produced by these bacteria are much less bioavailable, they are difficult to efficiently degrade by bacterial enzymes into smaller molecular weights (Mühlenbruch et al., 2018), which inhibits their sea-to-air transfer.



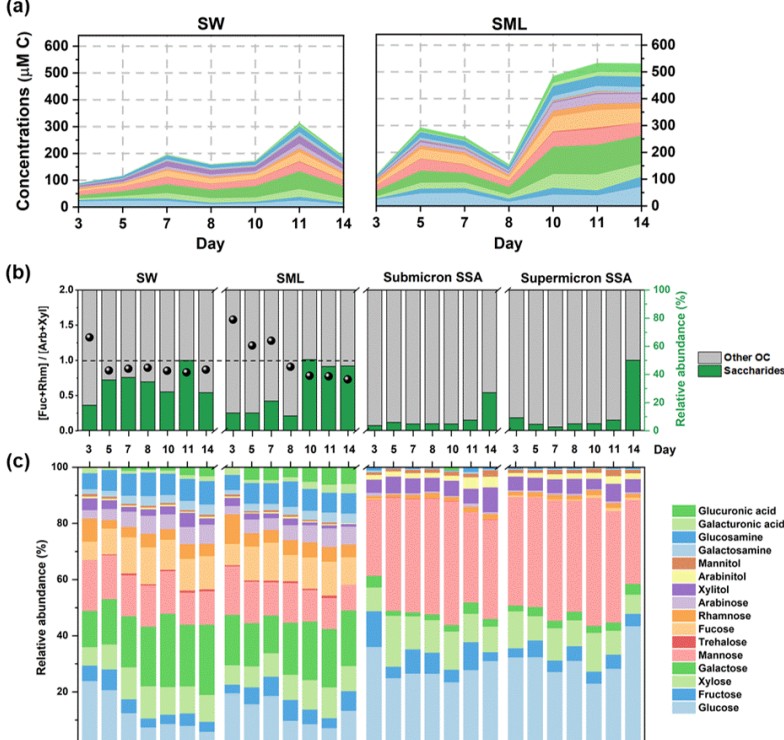

**Figure 6.** Sea-to-air transfer of saccharides. (a) Relative abundance of total saccharides in DOC. The bacterial activity is expressed as the ratio of the sum of the concentrations of fucose and rhamnose to the sum of the concentrations of arabinose and xylose (the black scatters). The activity is typically considered higher when the ratio is less than 1. (b) Composition and relative abundance of saccharides.

## 4 Atmospheric implications

Phytoplankton blooms are "pulse events" in ocean-atmosphere organic carbon cycle, transferring functionally specific molecules through SSA to the atmosphere, and potentially influencing cloud condensation nucleation and the atmospheric chemistry of SSA. In this study, we found that DOC enrichment in SSA during phytoplankton blooms can increase by up to 10-30 times, and this promoting effect resulted from the coupling of biological activity and "DOC-Bubble-SSA" interactions (Fig. 7). First, biological activity-driven organic carbon cycling in seawater triggers fluctuations in DOC concentration and composition. The dynamic accumulation of DOC directly influences SSA formation by altering the physical properties of seawater surfaces and indirectly impacts the sea-to-air transfer of DOC via SSA. Secondly, due to the varying enrichment capacities of different organic components in SSA, the changes in DOC composition and concentration in seawater caused by phytoplankton blooms are a key factor in the time-series fluctuations of the DOC enrichment factors. Finally, DOC produced



by phytoplankton cannot immediately transfer to SSA and often requires modification by heterotrophic microorganisms before

effective transfer. Note that size constraints on operationally defined DOC (<0.45 μm) exist for the enrichment both in film

drops and jet drops, with more stringent size constraints for film drops. Overall, proteins appear to be the main contributors to

DOC sea-to-air transfer during the phytoplankton bloom, while the contribution of saccharides increased significantly only

towards the end of the bloom.

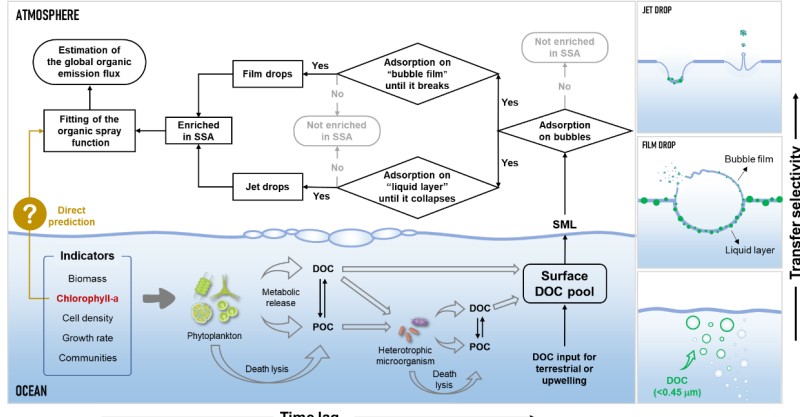

**Figure 7.** Sea-to-air transfer patterns of DOC via SSA during phytoplankton blooms.

Chl-a concentration, as readily available from satellite data, can indicate phytoplankton abundance in the surface layer of

the global oceans. However, current studies remain divided on whether Chl-a concentration can effectively predict the flux of

organic matter emissions via SSA. Our results suggest that the sea-to-air transfer of DOC is influenced by biological cycling

and its selective fractionation at different air-water interfaces, which may cause the non-significant correlation with Chl-a

levels during phytoplankton bloom. In fact, in more prevalent oligotrophic seas, the suppressed phytoplankton activity may

lead to a complete decoupling between Chl-a and sea-to-air transfer of DOC (Quinn et al., 2014). Therefore, predicting organic

spray emission fluxes based solely on Chl-a may introduce large uncertainties. Compared to previous studies, our research

offers new insights into the systematic and complex relationship between marine biological activities and SSA formation,

which might be fully considered in the sea-to-air transfer of DOC.

**Data availability**

The data in our study are available from the corresponding author on reasonable request.

**Supplement**

Additional experimental details, materials and methods, including description of experimental apparatus, sample collection

and detection, SSA morphology and chemical composition, and calculation methods and results.




**Author Contributions**

J. H., L. D. and J. L. designed and conceived the research; J. H. and F. X. conducted the experiments; J. H. performed data analysis; J. H. and J. L. wrote the original draft paper; N. T. T., C. G., M. H. and L. D. helped to write, review, and edit the manuscript.

**Competing interests.**

The contact author has declared that none of the authors has any competing interests.

**Acknowledgements**

We would like to thank Xiaoju Li from State Key laboratory of Microbial Technology of Shandong University for help and guidance in TEM.

**Disclaimer**

Publisher's note: Copernicus Publications remains neutral with regard to jurisdictional claims made in the text, published maps, institutional affiliations, or any other geographical representation in this paper. While Copernicus Publications makes every effort to include appropriate place names, the final responsibility lies with the authors.

**Financial support**

This work was supported by National Natural Science Foundation of China (22376121, 22361162668 and 42476033), National Key Research and Development Program of China (2023YFC3706203), and Intramural Joint Program Fund of State Key Laboratory of Microbial Technology (SKLMTIJP-2025-02).

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
