# Peer review of "Sea spray promotes the sea-to-air transfer of dissolved organic carbon during phytoplankton bloom"

_EGUsphere, 2025_

## Author Comment (AC1)

**Response to the comments from Community Comment 1**

We sincerely thank Ian Jenkinson for his valuable comments. Our manuscript has been revised according to the comments and our responses to the comments are as follows. For clarity, the comments are reproduced in blue, authors' responses are in black and changes in the manuscript are in red.

**A very interesting and important paper!**

However, quoting Lines 190--219, "Several studies have found that phytoplankton blooms can result in the formation of mucus on the water surface, which is typically an excessive accumulation of extracellular polysaccharides (Ternon et al., 2024; Medina-Pérez et al., 2021). In contract, this can increase the viscosity of SML and potentially enhance its surface tension (Jenkinson and Sun, 2010). From day 1 to day 5, the rapid increase in the surface tension of SML samples appears to be related to the rise in their saccharide concentration (see in Fig. 6a)."

As the authors say, mucus, secreted by organisms such as phytoplankton, consists of polymers can indeed increase viscosity of seawater. However, it tends to reduce surface tension below the value for "pure" (i. e. organics-free) seawater about 74 mN.m-1, not enhance (increase) it. As shown in the authors' Fig. 6d, the surface tension of SML water remained consistently less than that of subsurface water (SSW) by about 0.5 to 1 mN.m-1, consistent with enrichment in the SML. The much lower values at the beginning of the experiment remain enigmatic to me, unless they might have been caused by some tiny contaminant by a surfactant molecule such as detergent, often present on the surface of new apparatus. The subsequent increase could then have represented the incorporation of such a surfactant into other organic matter in the experiment, or its conversion or utilization by organisms present. I think this small issue does not affect the validity of the rest of the presentation.

**Author Reply**

Thank you for your comment. We agree with your view that, compared to pure seawater, the presence of organic matter generally reduces surface tension. Since the sea surface microlayer (SML) typically contains higher concentrations of organic matter than subsurface water (SSW), the surface tension of SML was consistently lower than that of SSW in our experiments.

We can rule out the possibility of these compounds originating from our homemade SSA simulation tank. On the one hand, the new equipment was thoroughly scrubbed with a brush and rinsed multiple times with both tap and deionized water before the next SSA experiment. Even if contaminants remained, they are unlikely to have such a significant impact on the surface tension of SML. On the other hand, the surface tension of SML showed a gradual increase over the first five days. If these organic compounds originated from the equipment surfaces, we would expect the surface tension of the SML to rapidly return to its "original" state during the second SSA experiment.

Another possibility is that the surface-active organic matter may already exist in the coastal seawater's microlayer. To investigate this further, we analyzed the mass spectrometry data. Figure 1 shows the base peak chromatograms for three samples: SSW and SML sample on Day 1 and SML sample on Day 9. We observed a prominent peak (Peak 1) between elution times of 21.1 and 21.9 minutes in the SML on Day 1, significantly higher than in the seawater on Day 1 and SML on Day 9. Results in Figure 2(a) indicate that the ion at m/z 221.0813 is the primary contributor to Peak 1, with an assigned molecular formula of C12H13O4 (error = 1.1 ppm). The results in Figure 2(b) show that the primary signal intensities in the secondary mass spectrometry fragments of the ion at m/z 221.0813 originate from m/z 177.0913 and 144.0964. These characteristic ions match those observed in the mass spectrum of diethyl phthalate (DEP) standard in the NIST Standard Reference Database (Figure 3). Therefore, Peak 1 can be primarily attributed to DEP. DEP is a commonly used plasticizer, and high concentrations (in the range of mg L-1 or mg kg-1) have been detected in various aquatic environments (Gani and Kazmi, 2016; Lu et al., 2023; Liang et al., 2024). Figure 1

reveals that high concentrations of DEP signal was present only in the Day 1 SML samples, while signals in the Day 1 seawater samples were very low. This could be due to DEP's low solubility in water and hydrophobic nature, which makes it significantly enriched in the SML. The DEP signal in the Day 9 SML sample was also low, likely due to reduced concentrations from biosorption or transformation processes (Gao and Chi, 2015; Liang et al., 2024). We further examined the relationship between DEP concentration and surface tension in artificial seawater (Figure 4). Even at extremely low concentrations, DEP can significantly reduce surface tension. For example, a DEP concentration of 2  $\mu$ M can reduce surface tension to the initial SML value of 65.84  $\pm$  0.36 mN m-1, which is significantly lower than DOC concentration in the SML at that time. Therefore, the presence of DEP in the SML at the start of the experiment was a significant factor contributing to its low surface tension.

We consider that DEP was present in the seawater from the outset of the experiment, likely originating from coastal pollution or being introduced during seawater transport. SML consists of an extremely thin layer at the water's surface, ranging from 1 to 1000 µm, occupying a negligible fraction of the total seawater volume. Although DEP exhibited strong mass spectrometry signals in the day 1 SML sample, its signal in SSW sample on Day 1 were very low (Figure 1). This suggests that the concentrations of DEP in the seawater used in our study were actually quite low. As a result, we no longer consider its impact on phytoplankton blooms.

Figure 1. Base peak chromatogram for three samples: SML sample on day 1(blue line), seawater sample on day 1(red line), SML sample on day 9 (black line).

Figure 2. The primary contributing ion of Peak 1 and its secondary mass spectrometry fragments.

Figure 3. Standard spectrum of diethyl phthalate from NIST Standard Reference Database 69: NIST Chemistry WebBook (<a href="https://webbook.nist.gov/chemistry">https://webbook.nist.gov/chemistry</a>). Note that the standard spectrum employs electron ionization, whereas we utilize an electrospray ionization source. Nevertheless, certain characteristic ions from the standard spectrum remain useful for our identification.

Figure 4. The relationship between different concentrations of DEP and the surface tension of artificial seawater.

We have revised the previous description in the manuscript.

**Page 7, lines 186-196**

As shown in Figure 2d, the surface tension of SML at the start of the experiment was measured to be  $65.84 \pm 0.36$  mN m-1, which exceeded our expectations. Using both primary and secondary mass spectrometry, we detected diethyl phthalate in the SML on Day 1. As common plasticizer, it is often found in coastal seawater and accumulates in SML due to its low solubility and hydrophobic nature (Lu et al., 2023), significantly reducing surface tension even at low concentrations (Figure S8). However, no diethyl phthalate was detected in bulk seawater on Day 1, which suggests that they likely do not influence phytoplankton blooms in bulk seawater. Detailed mass spectrometry analysis can be found in the Supplement. The surface tension of the SML increased rapidly from Day 1 to Day 5, possibly due to the rapid increase in DOC concentration in the SML during phytoplankton growth (Figure 3). Organic matter secreted by

microorganisms can significantly affect the physical properties of the SML (Jenkinson and Sun, 2010; Ternon et al., 2024), which may partially mitigate the low surface tension observed at the beginning. Furthermore, the rapid increase in surface tension may also be linked to the biosorption of diethyl phthalate or transformation by marine microorganisms (Liang et al., 2024; Gao and Chi, 2015).

**We added the following in the Supplement.**

**S5. Identification of phthalate esters in initial SML samples through mass spectrometry**

Figure S7(a) shows the base peak chromatograms for three samples: SSW and SML on Day 1 and SML on Day 9. We observed a prominent peak (Peak 1) between elution times of 21.1 and 21.9 minutes in the SML on Day 1, significantly higher than in the seawater on Day 1 and SML on Day 9. Results in Figure S7(b) indicate that the ion at m/z 221.0813 is the primary contributor to Peak 1, with an assigned molecular formula of  $C_{12}H_{13}O_4$  (error = 1.1 ppm). The results in Figure S7(c) show that the primary signal intensities in the secondary mass spectrometry fragments of the ion at m/z 221.0813 originate from m/z 177.0913 and 144.0964. These characteristic ions match those observed in the mass spectrum of diethyl phthalate (DEP) standard in the NIST Standard Reference Database (Figure S7(d)). Therefore, Peak 1 can be primarily attributed to DEP. DEP is a commonly used plasticizer, and high concentrations (in the range of mg L-1 or mg kg-1) have been detected in various aquatic environments (Gani and Kazmi, 2016; Lu et al., 2023; Liang et al., 2024). Figure 1 reveals that high concentrations of DEP signal was present only in the Day 1 SML, while signals in the Day 1 seawater were very low. This could be due to DEP's low solubility in water and hydrophobic nature, which makes it significantly enriched in SML. The DEP signal in the Day 9 SML was also low, likely due to reduced concentrations from biosorption or transformation processes (Gao and Chi, 2015; Liang et al., 2024). We further examined the relationship between DEP concentration and surface tension in artificial seawater (Figure S8). Even at extremely low concentrations, DEP can significantly reduce surface tension. For example, a DEP concentration of 2 µM can reduce surface tension to the initial SML value of  $65.84 \pm 0.36$  mN m-1, which is significantly lower than DOC concentration in the SML at that time. Therefore, the presence of DEP in the SML at the start of the experiment was a significant factor contributing to its low surface tension. Therefore, the presence of DEP in the SML sample at the start of the experiment was a significant factor contributing to its low surface tension.

Figure S7. Identification of phthalate esters in initial SML samples through mass spectrometry. (a) Base peak chromatogram for three samples: SML on Day 1(blue line), seawater on Day 1(red line), SML on Day 9 (black line); (b) Primary contributing ion of Peak 1 and its secondary mass spectrometry fragments; (c) Standard spectrum of diethyl phthalate from NIST Standard Reference Database 69: NIST Chemistry WebBook (<a href="https://webbook.nist.gov/chemistry">https://webbook.nist.gov/chemistry</a>). Note that the standard spectrum employs electron ionization, whereas we utilize an electrospray ionization source. Nevertheless, certain characteristic ions from the standard spectrum remain useful for our identification.

Figure S8. Relationship between different concentrations of DEP and the surface tension of artificial seawater.

**References**

Gani, K. M. and Kazmi, A. A.: Phthalate contamination in aquatic environment: A critical review of the process factors that influence their removal in conventional

- and advanced wastewater treatment, Critical Reviews in Environmental Science and Technology, 46, 1402-1439, https://doi.org/10.1080/10643389.2016.1245552, 2016.
- Gao, J. and Chi, J.: Biodegradation of phthalate acid esters by different marine microalgal species, Mar. Pollut. Bull., 99, 70-75, https://doi.org/10.1016/j.marpolbul.2015.07.061, 2015.
- Jenkinson, I. R. and Sun, J.: Rheological properties of natural waters with regard to plankton thin layers. A short review, J. Mar. Syst., 83, 287-297, <a href="https://doi.org/10.1016/j.jmarsys.2010.04.004">https://doi.org/10.1016/j.jmarsys.2010.04.004</a>, 2010.
- Liang, J., Ji, X., Feng, X., Su, P., Xu, W., Zhang, Q., Ren, Z., Li, Y., Zhu, Q., Qu, G., and Liu, R.: Phthalate acid esters: A review of aquatic environmental occurrence and their interactions with plants, J. Hazard. Mater., 470, 134187, <a href="https://doi.org/10.1016/j.jhazmat.2024.134187">https://doi.org/10.1016/j.jhazmat.2024.134187</a>, 2024.
- Lu, M., Jones, S., McKinney, M., Kandow, A., Donahoe, R., Cobb Faulk, B., Chen, S., and Lu, Y.: Assessment of phthalic acid esters plasticizers in sediments of coastal Alabama, USA: Occurrence, source, and ecological risk, Sci. Total Environ., 897, 165345, https://doi.org/10.1016/j.scitotenv.2023.165345, 2023.
- Ternon, E., Dinasquet, J., Cancelada, L., Rico, B., Moore, A., Trytten, E., Prather, K. A., Gerwick, W. H., and Lemée, R.: Sea-Air Transfer of Ostreopsis Phycotoxins Is Driven by the Chemical Diversity of the Particulate Fraction in the Surface Microlayer, Environ. Sci. Technol., 58, 18969-18979, <a href="https://doi.org/10.1021/acs.est.4c06691">https://doi.org/10.1021/acs.est.4c06691</a>, 2024.

---

## Author Comment (AC2)

**Reply to the comments from Anonymous Referee #1**

We deeply appreciate Anonymous Referee #1 for the thorough review of our manuscript. Our manuscript has been revised according to the comments and our responses to the comments are as follows. For clarity, the comments are reproduced in blue, authors' responses are in black and changes in the manuscript are in red color text.

The aim of the paper is to highlight relationships between biological activity, DOC concentration in seawater/sea surface microlayer and sea spray aerosols.

I have several criticisms that are important to address before considering resubmitting this paper:

1. Except for the introduction, this paper is difficult to read, it is extremely long and the rationale of the study is not easy to grasp. The paper feels like adding a result to another without deep interpretation of the data. While presenting and discussing their results, the authors should make the reader understand why this experiment was performed? What was the main question? How does it relate to great questions in this field? What are the main advances obtained from this experiment? A huge effort of restructuration of the results and discussion part is needed.

**Author Reply**

We have carefully addressed the issues raised by the Referee by clarifying the rationale of the study, by expanding the results and discussion and by highlighting the major achievements of this study. Furthermore, we deleted unnecessary discussions and the excessive text throughout. Specifically, the following were explored:

(1) The final paragraph of the introduction presents the experimental workflow of this study and elaborates on how the molecular formula of dissolved organic carbon guides our subsequent investigations into proteins, saccharides, and humic substances.

**Pages 2-3, lines 64-69**

This study explored the implications of sea-to-air transfer of DOC during phytoplankton blooms. Firstly, the macroscopic effects of phytoplankton blooms on

SSA formation and DOC enrichment were examined. Secondly, we employed high-resolution mass spectrometry to analyze the molecular profiles of DOC at different stages of sea-to-air transfer. Finally, by focusing on the most significant contributors, such as proteins, saccharides, and humic substances, the patterns of DOC sea-to-air transfer during phytoplankton blooms were investigated through a micro-to-macro approach.

(2) In Section 3.2 "Effects of DOC Variations on SSA Formation", we have incorporated comparisons with previous research findings and added the atmospheric meaning of DOC fluctuations inducing changes in SSA particle size.

**Page 8, lines 230-244**

Previous studies have explored the relationship between surface tension and the formation of SSA by adding surfactants at varying concentrations (Sellegri et al., 2006; Tyree et al., 2007; Song et al., 2024). Our results are consistent with the findings of these studies, suggesting that increased surface tension is a key factor that contributes to decreasing the number concentration of SSA while increasing the geometric mean diameter. It was reported that increased surface tension tends to inhibit the instability of bubble film edge and the development capillary waves during bubble bursting, thereby reducing the number of film drops and increasing the droplet size (Wang and Liu, 2025; Lhuissier and Villermaux, 2012). High surface tension strengthens the attraction between liquid molecules at the surface, enabling the bubble film to withstand the pressure difference between the inside and outside of the bubble (Wu et al., 2021; Wu et al., 2022). This resulted in an extended bursting time of the bubbles population, leading to the appearance of a long-lasting foam layer on the water surface from Day 8 to Day 18 (Fig. S2), which inhibited the production of SSA. During phytoplankton blooms, surface tension is no longer solely influenced by the concentration of a single species, but by fluctuations in the composition and concentration of DOC driven by biological activity. In coastal waters where surfactants are abundant, surface tension changes are similar to those observed in SML during phytoplankton blooms may be widespread. Biologically induced DOC fluctuations will directly affect the particle size

distribution of SSA, ultimately affecting SSA behavior such as atmospheric residence time and wet/dry deposition (Veron, 2015).

(3) In Section 3.3 "The Phytoplankton Bloom Promotes DOC Enrichment in SSA", we elucidate that variations in DOC enrichment factors within SSA may have a profound impact on its climatic effects. This further leads us to focus our subsequent discussion on specific DOC species.

**Page 10, lines 269-278**

Compared to particle size distribution, the significant variation in DOC's EF in SSA may have more profound implications for SSA's climate effect. First, from a morphological perspective, dried SSA particles consistently exhibit an organic shell enveloping an inorganic salt core (Figure S3). Since organic matter generally has lower hygroscopicity than sea salt, the presence of an organic shell suppresses the hygroscopic growth of SSA particles, thereby affecting their cloud condensation nucleus activity. The extent of this suppression depends on the type and concentration of organic matter (Bates et al., 2020; Lee et al., 2020; Cravigan et al., 2020). Second, biogenic DOC is essential in influencing the ice nucleation activity of SSA, particularly polysaccharides and proteins, which serve as effective ice nucleation sites promoting ice crystal formation (Pandey et al.; Hartmann et al., 2025). Finally, the presence of DOC also alters SSA's radiative effects and atmospheric reactivity (Bertram et al., 2018). Based on the above, we conclude that the EF of DOC in SSA provides a macroscopic description, which requires more detailed efforts to elucidate the sea-to-air transfer pattern of different organic species.

(4) In Section 3.5, we highlighted that proteins, saccharides, and humic compounds identified through mass spectrometry may be the primary contributors to DOC during phytoplankton blooms.

**Page 11, lines 309-316**

As illustrated in Fig. 4c, lipid-like, protein-like, carbohydrate-like and lignin-like molecules accounted for the majority of organic molecules transferred from sea to air.

Previous studies have confirmed that the DOC produced by algae consists of two major aliphatic groups: proteins and saccharides (Suo et al., 2024). Lignin-like molecules are widely considered to predominantly contribute to humic substances (Kim et al., 2003; Labeeuw et al., 2015). However, considering the effects of ionization mode and ionization efficiency, mass spectrometry results cannot directly reflect the relative abundance or concentration changes of specific DOC species. Therefore, additional methods were used to quantify the concentration fluctuations of protein, saccharides and humic substances in DOC to better understand the link between DOC's sea-to-air transfer and biological activity.

(5) In Section 4 "Atmospheric Implications", we further expanded on the climate effects of saccharides and proteins in dissolved organic carbon on SSA.

**Page 18, lines 454-4457**

Polysaccharides and amino acids produced by phytoplankton have been demonstrated to be key substances for efficient ice nucleation activity and are frequently detected in SSA and low-level clouds (Triesch et al., 2021a; Triesch et al., 2021b; Hartmann et al., 2025). Therefore, given the frequent occurrence of phytoplankton blooms and the enhancing effect of ocean warming, they will ultimately exert a profound influence on climate through the sea spray process.

2. Together with the restructuration, the authors should consider reviewing deeper the literature to support their experimental evidences. At many places, citations are missing. We sometimes don't know if the results presented are from the author's work or from the literature.

**Author Reply**

We have re-examined the section of "Results and Discussion" and when necessary, we re-phrased the initial text or supplemented references in several places to enhance the distinctiveness of our findings from other studies.

**Page 7, line 191-196**

Biermann et al. found that the decline in DOC concentration during the early

stages of phytoplankton blooms typically ceases after the depletion of inorganic nitrogen and phosphorus in seawater (Biermann et al., 2014). It has also been reported that the addition of inorganic nutrient not only promotes the heterotrophic consumption of DOC by phytoplankton blooms (Thornton, 2014), but also enhances bacterial production and respiration rates, thereby increasing their ability to utilize DOC (Carlson et al., 2004; Jiao et al., 2010; Cai and Jiao, 2008).

**Page 7, line 206-208**

The trend of SSA number concentration closely followed that of submicron SSA, as submicron SSA mainly contributes to the number concentration (Quinn et al., 2015).

**Page 8, lines 230-231**

Previous studies have explored the relationship between surface tension and the formation of SSA by adding surfactants at varying concentrations (Sellegri et al., 2006; Tyree et al., 2007; Song et al., 2024).

**Page 8, lines 242-244**

Biologically induced DOC fluctuations will directly affect the particle size distribution of SSA, ultimately affecting SSA behavior such as atmospheric residence time and wet/dry deposition (Veron, 2015).

**Page 10, lines 263-268**

Compared to supermicron SSA, the EF of DOC in submicron SSA consistently exhibited higher values and faster increases, which may be attributed to differences in SSA formation mechanisms. Before the bubble film ruptures at the water surface, the gravity continuously expels the liquid within it, while surface-active substances, being lighter, are pushed upward, forming a vanishingly thin film (Lhuissier and Villermaux, 2012). The resulting film drops are thus enriched with a higher concentration of organic matter. In contrast, jet drops primarily originate from the liquid at the air-water interface inside the bubble and are typically less enriched in organic matter than film drops (Crocker et al., 2022).

**Page 10, lines 269-278**

Compared to particle size distribution, the significant variation in DOC's EF in SSA may have more profound implications for SSA's climate effect. First, from a

morphological perspective, dried SSA particles consistently exhibit an organic shell enveloping an inorganic salt core (Figure S3). Since organic matter generally has lower hygroscopicity than sea salt, the presence of an organic shell suppresses the hygroscopic growth of SSA particles, thereby affecting their cloud condensation nucleus activity. The extent of this suppression depends on the type and concentration of organic matter (Bates et al., 2020; Lee et al., 2020; Cravigan et al., 2020). Second, biogenic DOC is essential in influencing the ice nucleation activity of SSA, particularly polysaccharides and proteins, which serve as effective ice nucleation sites promoting ice crystal formation (Pandey et al.; Hartmann et al., 2025). Finally, the presence of DOC also alters SSA's radiative effects and atmospheric reactivity (Bertram et al., 2018). Based on the above, we conclude that the EF of DOC in SSA provides a macroscopic description, which requires more detailed efforts to elucidate the sea-to-air transfer pattern of different organic species.

**Page 11, lines 299-306**

This suggests that the composition of DOC in seawater is influenced by biological activity during phytoplankton blooms (Meon and Kirchman, 2001), which in turn affects the sea-to-air transfer of DOC via SSA (Schmitt-Kopplin et al., 2012). For instance, our results show that the proportion of shared organic molecular formulas in SW, SML, submicron SSA, and supermicron SSA increased from 12.4% on Day 1 to 16.2% on Day 9 and 26.3% on Day 18.

**Page 15, lines 371-373**

Phytoplankton typically sequesters the excess carbon as saccharides in energy storage materials, cell walls, and extracellular polysaccharides, and this process is influenced by phytoplankton growth, heterotrophic bacteria, and environmental conditions (Mühlenbruch et al., 2018).

**Page 15, lines 387-388**

Seawater and SML exhibit different time series of bacterial activity, suggesting that they may harbor distinct microbial communities (Rahlff et al., 2023; Rahlff et al., 2019; Reinthaler et al., 2008).

**Page 18, lines 454-456**

Polysaccharides and amino acids produced by phytoplankton have been demonstrated to be key substances for efficient ice nucleation activity and are frequently detected in SSA and low-level clouds (Triesch et al., 2021a; Triesch et al., 2021b; Hartmann et al., 2025).

3. There is a lot of speculation in the results and discussion part, especially on the biological side of the experiment. The authors are sometimes over interpretating the data, they should restrict their discussion to what can effectively be discussed (not the biology since the chla was the only biological parameter measured, this gives no idea of what happened in the MART).

**Author Reply**

We have restricted our interpretation of the biological processes occurring in MART to avoid overinterpretation. The revised texts are as follows:

**Page 7, lines 191-196**

Biermann et al. found that the decline in DOC concentration during the early stages of phytoplankton blooms typically ceases after the depletion of inorganic nitrogen and phosphorus in seawater (Biermann et al., 2014). It has also been reported that the addition of inorganic nutrient not only promotes the heterotrophic consumption of DOC by phytoplankton blooms (Thornton, 2014), but also enhances bacterial production and respiration rates, thereby increasing their ability to utilize DOC(Carlson et al., 2004; Jiao et al., 2010; Cai and Jiao, 2008).

**Page 8, lines 224-229**

The surface tension of the SML increased rapidly from Day 1 to Day 5, possibly due to the rapid increase in DOC concentration in the SML during phytoplankton growth (Figure 3). Organic matter produced by microorganisms can significantly affect the physical properties of the SML (Jenkinson and Sun, 2010; Ternon et al., 2024), which may partially mitigate the low surface tension observed at the beginning.

**Page 15, lines 371-373**

Phytoplankton typically sequesters excess the carbon as saccharides in energy

storage materials, cell walls, and extracellular polysaccharides, and this process is influenced by phytoplankton growth, heterotrophic bacteria, and environmental conditions (Mühlenbruch et al., 2018).

**Page 15, lines 387-388**

Seawater and SML exhibit different time series of bacterial activity, suggesting that they may harbor distinct microbial communities (Rahlff et al., 2023; Rahlff et al., 2019; Reinthaler et al., 2008).

**Page 15, lines 392-395**

However, the percentage of saccharides in SSA remained consistently below 10%, with a significant increase only observed on Day 14, corresponding to the peak bacterial activity in the SML. Hasenecz et al. found that addition of heterotrophic bacteria significantly increased the saccharide enrichment in SSA, as the enzymes released by these bacteria further modified the saccharides (Hasenecz et al., 2020).

**Page 16, lines 422-426**

For example, our experimental results showed that fucose was scarcely detected in submicron SSA, but was only enriched in the SML and supermicron SSA. Previous studies indicate that fucose-constituted polysaccharides primarily range from 50 nm ( $\approx$  100 kDa) to 450 nm, representing a relatively large size (Jayarathne et al., 2022). It was also suggested that these polysaccharides resist bacterial hydrolysis (Murray et al., 2007). This may explain why these polysaccharides did not effectively enter the submicron SSA during our experimental period.

4. Regarding the experimental scheme, my main criticism is the method used to get a phytoplankton bloom. The carboys filled with SW were left at the sun for 18 days, and the temperature inside was never measured. Were they even ventilated? The expected temperature increase could have drastically modified the microbial community with strong pH modification. The first incubation is probably involving larger phytoplankton communities while the others most likely smaller communities and lots of bacteria. Were there any microscope observations made over the course of the experiment? It has been shown that the physiology of phytoplankton can influence the formation of SSA:

species, growth phase... This is only acknowledged once in the discussion (page 14, line 338).

**Author Reply**

Our phytoplankton bloom experiment was conducted from June 1 to June 18, 2024, on an open plot at Shandong University's Qingdao Campus (120.685°E, 36.364°N) (Figure 2a). During this period, the local outdoor temperature remained relatively stable, varying between  $19.39 \pm 2.06$  °C and  $22.98 \pm 1.82$  °C (Figure 2b). The temperature variation of seawater inside the containers primarily depends on air heat conduction and direct solar heating. Considering that seawater has a specific capacity approximately 3000 times that of air and a large volume ( $\approx$ 30 L), we speculate that the diurnal fluctuation in seawater temperature within the containers may not exceed the diurnal fluctuation in air temperature, which is  $3.59 \pm 1.21$  °C. The fluctuations in seawater temperature during our phytoplankton blooms differ from actual conditions, as records show that the diurnal fluctuation in surface seawater temperature at a coastal station in Qingdao (120.302°E, 36.052°N) from June 1 to 18, 2019, was  $1.06 \pm 0.42$  °C (Figure 2c) (Cao et al., 2024). Although these temperature variations could influence the development of phytoplankton blooms and seawater pH, it is unlikely that they could undermine the conclusions. During the experiment, instead of the ventilation, the containers at were shaken at least three times daily to promote better mixing of the seawater and the growth of phytoplankton growth.

Our experimental design focused primarily on the impact of phytoplankton blooms on the sea-to-air transfer of dissolved organic carbon via SSA. During the experiment, phytoplankton biomass was characterized using chlorophyll a concentration, while bacterial activity in seawater was assessed via the concentration ratio [fucose + rhamnose] / [arabinose + xylose] (Jayarathne et al., 2022). We explored potential conversion relationships between POC and DOC and used multiple methods to identify and quantify the sea-to-air transfer processes of the most abundant proteinaceous, saccharide, and humic substances in DOC. The experimental design and results presented in this manuscript adequately meet the research objectives. Nevertheless, we recognize that microscopic observations of changes in marine microbial communities

during different stages of phytoplankton blooms would enhance our understanding of the relationship between biological activity and DOC sea-to-air transfer. However, due to the scope of the research and resource constraints, this aspect was not included in the current experimental design. In our future studies, it will be fully considered and integrated.

Figure R1. (a) Location where phytoplankton bloom experiments were conducted; (b) Local air temperature; (c) Sea surface temperature along the Qingdao coast.

Finally, we clarified the discrepancy in seawater temperature between natural environmental conditions and our experiments during phytoplankton blooms, and elaborated on the associated limitations and potential implications.

**Page 3, lines 77-83**

The average diurnal fluctuation in local outdoor air temperature is 3.59 °C (temperature average:  $21.19 \pm 2.60$  °C). Seawater temperature fluctuations in the containers primarily depends on air heat conduction and direct solar heating. However, given seawater's high specific heat capacity, its diurnal variation is likely smaller than that of air temperature, with an average comparable to air temperature. This value is slightly higher than the 1.04 °C diurnal variation (temperature average: 18.72°C  $\pm$

1.02 °C) recorded for coastal seawater in Qingdao during the same period (Cao et al., 2024). Although this temperature discrepancy could influence the development of phytoplankton blooms, it is unlikely to significantly affect the conclusions.

The text format should be revised as many extra dots and extra space are present in the text.

**Author Reply**

We have revised the manuscript and removed extra dots and extra spaces from the revised manuscript.

Here are some examples in the text:

Section 3.1

We don't know the duration of the MART experiments

**Author Reply**

We have made the necessary revisions on page 3, lines 93-94.

All SSA generation experiments were conducted at a constant room temperature, typically starting around 9:00 AM and lasting for 8 to 9 hours.

Did you measure the N and P content to say they are depleted? How do you know?

**Author Reply**

We cited the experimental results from Biermann et al. (Biermann et al., 2014), who conducted mesoscale experiments using Baltic Sea water at two temperatures (T =  $2.7 \pm 0.3$  °C and T +  $6 = 8.3 \pm 0.3$  °C) under three light conditions (high light (HL ~ 8.9 mol quanta m-2 day-1), intermediate light (IL ~ mol quanta m-2 day-1), and low light (LL ~ mol quanta m-2 day-1)). In all six mesocosm setups, dissolved organic carbon (DOC) concentrations in seawater gradually decreased over the first 10 days. This decline ceased only after the depletion of inorganic phosphorus (DIP), inorganic nitrogen (DIN), and silicate, after which DOC concentrations began to rise again (Figure R2). The authors reported that DOC concentrations declined at the beginning of their experiment in all mesocosms from an initial overall average of 311  $\pm$  33  $\mu$ mol

 $L^{-1}$  to a minimum of  $239 \pm 5.2 \ \mu mol \ L^{-1}$  between Days 8 and 10, and started to increase again after exhaustion of DIP and DIN, and hence earlier in T + 6 treatments. This hypothesizes that the observed DOC decline in our experiments may be linked to the depletion of nitrogen and phosphorus nutrients in the seawater, despite these indicators have not been measured directly.

To further the potential association between the decline in DOC concentrations and N and P depletion in this study, we inserted the following in the revised manuscript.

**Page 6, lines 176-178**

Biermann et al. found that the decline in DOC concentration during the early stages of phytoplankton blooms typically ceases after the depletion of inorganic nitrogen and phosphorus in seawater (Biermann et al., 2014).

Figure R2. Six mesocosm experiments conducted by Biermann et al. (Biermann et al., 2014). The trends in dissolved organic carbon (DOC) concentrations in seawater are compared with those of inorganic phosphorus, inorganic nitrogen, and silicate.

**Phytoplankton may also use DOC**

**Author Reply**

We have made the necessary addition to the manuscript.

**Page 7, lines 193-196**

It has also been reported that the addition of inorganic nutrient not only promotes

the heterotrophic consumption of DOC by phytoplankton blooms (Thornton, 2014), but also enhances bacterial production and respiration rates, thereby increasing their ability to utilize DOC (Carlson et al., 2004; Jiao et al., 2010; Cai and Jiao, 2008).

**Line 165 – no measurement of bacterial activity**

**Author Reply**

The corresponding text has been deleted to remove any ambiguity.

**Section 3.2**

Why presenting the chla first and not presenting the DOC right away as it is the aim of this section?

**Author Reply**

The original sentence has been modified in the revised manuscript to link Chl-a concentration to SSA particle size distribution.

**Page 6, lines 205-206**

Prior to Day 10, the production of submicron SSA increased and then decreased while supermicron SSA exhibited an opposite trend.

**Citation expected line 179**

**Author Reply**

We have added the citation.

**Page 8, line 210-211**

Prior to Day 10, the production of submicron SSA increased and then decreased, while supermicron SSA exhibited an opposite trend. The trend of SSA number concentration closely followed that of submicron SSA, as submicron SSA mainly contributes to the number concentration (Quinn et al., 2015).

**Line 180: when? At each experiment?**

**Author Reply**

To further clarify our original idea, we have rewritten the sentence as:

**Page 8, lines 208-209**

During the phytoplankton bloom, the geometric mean diameter of SSA increased from  $103.8 \pm 5.0$  nm on Day 1 to  $136.3 \pm 5.4$  nm on Day 10, before gradually decreasing to  $115.0 \pm 6.9$  nm (Fig. 2c).

**Line 181: where is the data to support this?**

**Author Reply**

This statement represents one of our key points. It introduces the idea that fluctuations in DOC concentrations during phytoplankton blooms are a significant factor driving substantial changes in the formation of sea spray aerosol. In the original manuscript, we placed this statement at the end of the first paragraph. After careful consideration, we have moved it to the beginning of the second paragraph to minimize potential misunderstandings.

**Page 8, lines 205-212**

The distributions of SSA particle size during the phytoplankton bloom are shown in Fig. 2a-b. Prior to Day 10, the production of submicron SSA increased and then decreased while supermicron SSA exhibited an opposite trend. The trend of SSA number concentration closely followed that of submicron SSA, as submicron SSA mainly contributes to the number concentration (Quinn et al., 2015). During the phytoplankton bloom, the geometric mean diameter of SSA increased from  $103.8 \pm 5.0$  nm on Day 1 to  $136.3 \pm 5.4$  nm on Day 10, before gradually decreasing to  $115.0 \pm 6.9$  nm (Fig. 2c).

The dynamic accumulation of DOC during phytoplankton blooms may have a significant impact on bubble bursting and SSA formation by modifying seawater properties. As an important surface property, surface tension has been proven to be an influential parameter in controlling bubble bursting and SSA formation (Tammaro et al., 2021; Sellegri et al., 2006).

**Author Reply**

We have revised the title of Section 3.3.

3.3 Phytoplankton Bloom Promotes DOC Enrichment in SSA

Lines 218 - 224 = this is method and should be moved

**Author Reply**

We have moved the content to the Methods section.

Line 224: targeted analysis of which compounds?

**Author Reply**

We performed mass spectrometry analysis on the dissolved organic carbon (DOC) in SSA samples collected on Day 1, Day 9, and Day 18. This non-targeted analysis used the MFAssignR program to identify all molecular formulas and their intensities through steps such as noise control, isotope filtering, internal mass recalibration, and molecular formula assignment. MFAssignR is widely recognized for its effectiveness in performing consistent and efficient non-targeted analysis of complex environmental mixtures (Schum et al., 2020; Radoman et al., 2022). The assigned molecular formulas were then plotted on the Van Krevelen diagram (O/C on the *X*-axis, H/C on the *Y*-axis). By examining the distribution of different compound categories on the VK diagram (Suo et al., 2024), we assessed the contributions of various chemical classes to the DOC. Our findings revealed that proteins, saccharides, and humic compounds are significant contributors to DOC. Based on these results, we subsequently employed a fluorescence spectrometer to investigate the sea-to-air transfer of proteins and humic compounds, and an ion chromatograph to explore the transfer of saccharides.

A brief explanation to this was provided in the revised manuscript.

**Page 9, lines 254-256**

Due to the limited SSA collection, samples on Day 1, 9, and 18 were not analyzed for the EF of DOC, which were only used for mass spectrometry analysis to identify the primary DOC molecular types.

**Line 226 – is it from the literature or is it your result?**

**Author Reply**

This result stems from our own research. We have explained this in the manuscript.

**Page 9, lines 255-257**

In our phytoplankton bloom experiments, the EFs of DOC in SML, supermicron SSA, and submicron SSA increased by up to ~4-fold, 10-fold, and 30-fold, respectively.

Please check the literature on the effect of microalgal growth phases and formation of SSA. There is no discussion on the differences observed between EF for submicron and supermicron SSA. The authors could try to interpret this difference, especially after the peak of the bloom where submicron is produced. Were atmospheric reactions allowed to form nanoparticles?

**Author Reply**

We have incorporated a discussion on the differences in EF between submicron and supermicron SSA. Additionally, volatile gases released by phytoplankton, such as dimethyl sulfide, have been shown to form nanoparticles through atmospheric chemical reactions or by directly participating in microdroplet interface reactions to form organic sulfate esters (Jang et al., 2025). Organic aerosol particles formed by the secondary oxidation of isoprene released by phytoplankton (Meskhidze and Nenes, 2006) may also be present in our collected SSA samples. However, regarding these nanoparticles: first, the scanning electrical mobility size analysis we employed cannot detect particle number concentrations below 10 nm; second, even if these particles are sufficiently abundant, their extremely small size limits their overall impact; and finally, since our experiments did not focus on secondary organic aerosol formation, an effective discussion on this matter is beyond the scope of this study. This was clarified in the revised manuscript.

**Page 10, lines 263-268**

Compared to supermicron SSA, the EF of DOC in submicron SSA consistently exhibited higher values and faster increases, which may be attributed to differences in SSA formation mechanisms. Before the bubble film ruptures at the water surface,

gravity continuously expels the liquid within it, while surface-active substances, being lighter, are pushed upward, forming a vanishingly thin film (Lhuissier and Villermaux, 2012). The resulting film drops are thus enriched with a higher concentration of organic matter. In contrast, jet drops primarily originate from the liquid at the air-water interface inside the bubble and are typically less enriched in organic matter than film drops (Crocker et al., 2022).

**Section 3.4**

Line 238 What samples are considered here?

**Author Reply**

We have provided more detailed descriptions of the samples selected for mass spectrometry analysis in the manuscript.

**Page 11, lines 285-287**

To investigate the link between the sea-to-air transfer of DOC and biological activity, samples of submicron SSA, supermicron SSA, SML, and seawater were collected during the early (Day 1), peak (Day 9), and late (Day 18) stages of the phytoplankton bloom for mass spectrometry analysis.

**Line 257 this assumption should be put in perspective with other studies**

**Author Reply**

Appropriate citations have been included.

**Page 11, lines 302-304**

This suggests that the composition of DOC in seawater is influenced by biological activity during phytoplankton blooms (Meon and Kirchman, 2001), which in turn affects the sea-to-air transfer of DOC via SSA (Schmitt-Kopplin et al., 2012).

**Line 260: this is probably due to the biology in the MART**

**Author Reply**

Indeed, we acknowledge that the increase in the proportion of dissolved organic carbon molecular formulas shared among seawater, sea surface microlayer, submicron SSA, and supermicron SSA can be attributed to biological activity, which results from phytoplankton blooms. Our hypothesis is further supported by previous findings.

**Page 11, lines 301-306**

As shown in Figure 4a, the total intensity of molecular formulas assigned to seawater remained relatively stable across the three stages of the phytoplankton bloom, while the total number exhibited a stepwise decline. This suggests that the composition of DOC in seawater is influenced by biological activity during phytoplankton blooms (Meon and Kirchman, 2001), which in turn affects the sea-to-air transfer of DOC via SSA (Schmitt-Kopplin et al., 2012). For instance, our results show that the proportion of shared organic molecular formulas in SW, SML, submicron SSA, and supermicron SSA increased from 12.4% on Day 1 to 16.2% on Day 9 and 26.3% on Day 18.

**Section 3.5**

This is difficult section, What is HULIS? This should be explained in details and the authors should use another wording than HULIS 1, PRLIS, there are too many acronyms, it's difficult to remember at this stage of the paper without previous description and introduction.

**Author Reply**

HULIS refers to humic-like substances, where HULIS 1 denotes substances with excitation wavelengths < 245 nm or 320 nm and emission wavelengths at 396 nm, while HULIS 2 denotes substances with excitation wavelengths at 260 or 360 nm and emission wavelengths at 450–455 nm. PRLIS refers to protein-like substances with excitation wavelengths at 280 nm and emission wavelengths at 330 nm. We have restructured the final paragraph of the Introduction for a better flow in the following sections.

**Page 2, lines 64-69**

This study explored the implications of sea-to-air transfer of DOC during phytoplankton blooms. Firstly, the macroscopic effects of phytoplankton blooms on SSA formation and DOC enrichment were examined. Secondly, we employed high-resolution mass spectrometry to analyze the molecular profiles of DOC at different

stages of sea-to-air transfer. Finally, by focusing on the most significant contributors, such as proteins, saccharides, and humic substances, the patterns of DOC sea-to-air transfer during phytoplankton blooms were investigated through a micro-to-macro approach.

In Section 3.5, we provided further clarification on the definition of "HULIS."

**Page 13, lines 325-329**

The peaks of protein-like substances (PRLIS) are mainly at (280 nm)/ (330 nm), and most of them were due to tryptophan-like substances (Santander et al., 2022). The humic-like substances (HULIS) peaks mainly appear at the excitation/emission wavelengths of (<245 nm or 320 nm) / (396 nm) for HULIS 1, and at (260 or 360 nm) / (450-455 nm) for HULIS 2. The production of HULIS 1 is primarily linked to heterotrophic processes, whereas HULIS 2 is a photooxidation product and, as such, contains a higher oxygen content than HULIS 1 (Santander et al., 2023; Barsotti et al., 2016).

**Lines 280 - 283: this is method and should be moved**

**Author Reply**

Since this text describes the three fluorescent substances that coexist in seawater, sea surface microlayer, submicron SSA, and supermicron SSA, as revealed by the results of excitation-emission matrix combined with parallel factor, we believe it is at the appropriate place.

In the revised manuscript, we have further modified these sentences.

**Page 13, lines 323-329**

By EEM-PARAFAC method (Fig. 5a), three fluorescence compounds co-exiting in SW, SML, submicron SSA, and supermicron SSA were identified. The peaks of protein-like substances (PRLIS) are mainly at (280 nm)/ (330 nm), and most of them were due to tryptophan-like substances (Santander et al., 2022). The humic-like substances (HULIS) peaks mainly appear at the excitation/emission wavelengths of (<245 nm or 320 nm)/ (396 nm) for HULIS 1, and at (260 or 360 nm)/ (450-455 nm)

for HULIS 2. The production of HULIS 1 is primarily linked to heterotrophic processes, whereas HULIS 2 is a photooxidation product and, as such, contains a higher oxygen content than HULIS 1 (Santander et al., 2023; Barsotti et al., 2016).

Line 284: be more specific, what compounds are you talking about?

**Author Reply**

We have further explained this.

**Page 13, lines 330-331**

The EEM intensities of PRLIS, HULIS 1, and HULIS 2 in seawater, sea surface microlayer, submicron SSA, and supermicron SSA are shown in Figure 5b.

Line 291: not sure what's the conclusion of the authors here.

**Author Reply**

We have removed this conclusion and revised the sentence.

**Page 13, lines 336-337**

Due to their hydrophilic groups (-NH2 and -COOH) and hydrophobic carbon chains, they have been reported to exhibit strong enrichment potential in the SML and SSA (Triesch et al., 2021a; Triesch et al., 2021b).

Line 293: where is this match visible in the manuscript?

**Author Reply**

We have rewritten this conclusion to make the expression clearer.

**Page 13, lines 337-340**

As shown in Figure 5b, the EEM intensity of PRLIS in SSA rapidly peaks on Day 7 before declining. This pattern closely aligns with the DOC's enrichment factor trend in SSA presented in Figure 3, indicating that PRLIS was the primary contributor to the increase of DOC's EF during the phytoplankton bloom.

Line 300: this is vague regarding the results provided

**Author Reply**

Considering also the Referee's previous suggestions, we have moved the discussion regarding organic matter enrichment in submicron SSA and supermicron SSA to Section 3.3 "The Phytoplankton Bloom Promotes DOC Enrichment.", and a simplified discussion of the results is given as:

Page 13, lines 342-345

Compared to supermicron SSA, the EEM intensities of the three organic compounds are higher in submicron SSA. Consistent with previous studies, besides the properties of the organic matter itself, the sea-to-air transfer of DOC is also influenced by the different generation mechanism of SSA (Crocker et al., 2022).

Line 304-305: this is difficult to understand

**Author Reply**

We have restructured the relevant discussion.

Page 13, lines 347-351

We found that the EEM intensities of the three compounds in seawater are positively correlated with the DOC concentration, while in the SML, these compounds are significantly positively correlated with the POC concentration in seawater. This implies that DOC in the SML likely originates from POC in seawater rather than DOC. Within the same samples (seawater, SML, submicron SSA, or supermicron SSA), PRLIS, HULIS 1, and HULIS2 maintained significant or near-significant positive correlations; however, their correlation weakened across different samples (Fig. 5d).

Section 3.6

Line 328, 329 and 330: missing citations, are they results obtained in your study?

**Author Reply**

This is not our results and now, we have added the relevant citations.

Page 15, lines 371-373

Phytoplankton typically sequesters excess carbon as saccharides in energy storage materials, cell walls, and extracellular polysaccharides, and this process is influenced by phytoplankton growth, heterotrophic bacteria, and environmental conditions

**(Mühlenbruch et al., 2018).**

Line 340: how do you know about the degradation of saccharides; do you have data to support this assumption?

**Author Reply**

The hypothesis in question does not have direct empirical data from our experiments; we cited the a previous study to support this (Hasenecz et al., 2020). We removed this assumption in the revised manuscript to help reduce overinterpretation in our biological analyses.

Line 346: I doubt this study monitored the bacterial activity, they measured chla and saccharides. If they did then you can use their conclusion, if they suggested it, then it's not the conclusion of their study and cannot be used as such.

**Author Reply**

Upon verification, we confirm that the study by (Jayarathne et al., 2022) employed the molar ratio of (Fuc + Rha): (Ara + Xyl) to assess bacterial activity and concentration in seawater during phytoplankton blooms. To further validate whether this ratio effectively represents bacterial activity in seawater, we conducted a detailed examination. This molar ratio is commonly used to describe microbial metabolic pathways for saccharides and reflects the bacterial capacity to metabolize specific carbon sources. It holds particular significance in studies of bacterial growth, metabolic pathways, and environmental adaptation. Generally, (Fuc + Rha): (Ara + Xyl) serves as a key indicator for estimating microbial degradation levels, especially in research on recalcitrant dissolved organic matter degradation (Jiao et al., 2010). When the molar ratio falls below 1, it typically indicates environments characterized by high microbial (primarily bacterial) activity and extensively degraded humus, where microorganisms preferentially metabolize arabinose (Ara) and xylose (Xyl). In such conditions, fucose (Fuc) and rhamnose (Rha) are either present at lower concentrations or have been degraded (Engbrodt and Kattner, 2005; Frimmel, 1998). Therefore, a molar ratio below 1 can serve as an indirect biological indicator of the biodegradation state in the

environment.

Line 346: this has been shown in many papers, they should be cited here

**Author Reply**

We have supplemented the references.

Page 15, lines 387-388

Seawater and SML exhibit different time series of bacterial activity, suggesting that they may harbor distinct microbial communities (Rahlff et al., 2023; Rahlff et al., 2019; Reinthaler et al., 2008).

Other details

Section 2.1: medium F/2

**Author Reply**

We considered that using Guillard's F/2 medium might cause the phytoplankton bloom process to proceed too rapidly. Hence, we opted for Guillard's F/4 medium instead, at half the concentration of Guillard's F/2 medium.

We have revised the description in the manuscript.

Page 3, lines 74-75

Guillard's F/4 medium was added to each container, and these containers were placed outdoors on a flat to promote phytoplankton blooms under natural sunlight (Fig. S1).

Section 2.2: how long was each MART incubation? What time of the day were the seawater sampling made (this would influence the chemical composition of the phytoplankton community)?

**Author Reply**

We have added the duration of each MART experiment and the sampling times for seawater to the manuscript.

Page 3, line 93-94

All SSA generation experiments were conducted at a constant room temperature,

typically starting around 9:00 AM and lasting for 8 to 9 hours.

**Page 4, line 105-106**

Seawater samples were collected within the half-hour preceding the start of the nascent SSA generation.

How was chosen the extraction method of the SSA? Why using ultrapure water only and not other solvents?

**Author Reply**

Since the current research primarily focuses on the sea-to-air transfer of dissolved organic carbon, water was used to extract organic matter from SSA. We have provided an explanation in the manuscript.

**Page 4, lines 98-99**

Since this study focuses on the sea-to-air transfer of DOC, all SSA samples were extracted with ultrapure water (>18.2 M $\Omega$ ·cm, 25 °C, Millipore) and the extractions were filtered with 0.45  $\mu$ m filters.

**How was POC sampled?**

**Author Reply**

In Section 2.2, we have described the operation of filtering seawater using GF/F membranes in our original manuscript: "Seawater was collected at a depth of 10 cm in each container and immediately filtered at low pressure (≤0.2 MPa, avoiding the Chl-a loss) through a GF/F filter (47 mm, Whatman, UK)."

To clarify further, we emphasize in Section 2.3.2 that the POC measurements were conducted using the GF/F filters described in Section 2.2.

**Page 4, lines 123-125**

Quantitative measurements of POC and Chl-a concentrations in seawater were carried out using the GF/F filters described in Section 2.2. Specifically, the concentration of POC in seawater was determined using an elemental analyzer (Elementar, UNICUBE), which measured the POC content in a 1 cm diameter circular area on the GF/F filter.

**Section 2.3.1 - Chemical analysis of SSA or all chemical analysis?**

**Author Reply**

We are referring to all chemical analysis. The title of Section 2.3 has been changed from "Aerosol Characterization and Chemical Analysis" to "Characterization and Chemical Analysis".

**The methods should describe**

- sampling of seawater and bloom inducing

**Author Reply**

The description of seawater collection has been provided in Section 2.2 of the original manuscript: "Seawater was collected at a depth of 10 cm in each container and immediately filtered at low pressure (≤0.2 MPa, avoiding the Chl-a loss) through a GF/F filter (47 mm, Whatman, UK). Both filters and filtered seawater were stored at -20 °C in a dark environment."

Now, we provide a more detailed description of the triggering phytoplankton blooms, including nutrient additions and temperature changes during the period.

**Page 3, lines 72-83**

Seawater was collected on May 31, 2024, at Shazikou Pier (120°33'28" E, 36°6'37" N) Qingdao, China, and immediately transported to the laboratory, where it was filtered through a 1-mm mesh sieve and transferred into 30 transparent polycarbonate containers, each with a capacity of 28 liters. Guillard's F/4 medium was added to each container, and these containers were placed outdoors on a flat to promote phytoplankton blooms under natural sunlight (Fig. S1). The phytoplankton bloom experiment began on June 1st, 2024, and lasted for 18 days. During this period, 10 simulation experiments on nascent SSA were conducted. The average diurnal fluctuation in local outdoor air temperature is 3.59 °C (temperature average: 21.19 ± 2.60 °C). Seawater temperature fluctuations in the containers primarily depend on air heat conduction and direct solar heating. However, given seawater's high specific heat capacity, its diurnal variation is likely smaller than that of air temperature, with an

average comparable to air temperature. This value is slightly higher than the 1.04 °C diurnal variation (temperature average: 18.72°C  $\pm$  1.02 °C) recorded for coastal seawater in Qingdao during the same period (Cao et al., 2024). Although this temperature discrepancy could influence the development of phytoplankton blooms, it is unlikely to significantly affect the conclusions.

- experiment: time of sampling, sampling methods for all parameters, sub-sampling for different analysis and storing
- analysis of the different samples, each of the samples must have a specific name, "sample" is too vague and the reader doesn't know what sample the authors are considering

**Author Reply**

We thank the Referee for the valuable feedback. We have revised the experimental and analytical sections to provide clearer descriptions in line with the Referee's suggestions. Specifically, we have detailed the collection times for each parameter, the sampling methods employed, and the sub-sampling and storage procedures for the different analyses. To eliminate any ambiguity associated with the term "sample", we have assigned specific names to each sample and consistently label them throughout the manuscript, ensuring that readers can easily identify the samples being referenced.

Below, we outline the major revisions to the experimental methods section.

(1) The sampling time of nascent sea spray aerosol

**Page 3, lines 93-94**

All SSA generation experiments were conducted at a constant room temperature, typically starting around 9:00 AM and lasting for 8 to 9 hours.

(2) Drying, collection, particle size classification, and extraction of SSA.

**Page 4, lines 93-103**

More details on SSA generation are provided in the Supplement. Nascent SSA was transported with purified air (Zero Air Supply, Model 111, Thermo Scientific), and the airflow was dried to a relative humidity below 30% (Monotube Dryer, MD700-12F-3, Perma Pure, USA) before collection and measurement. At this relative humidity,

nascent SSA can become completely dry. Single particles of SSA were collected by a single particle sampler (DKL-2, Genstar electronic technology Co., Ltd., China) and then analyzed by transmission electron microscopy (TEM, FEI Tecnai G2 F20, Thermo Fisher Scientific, USA). Using a low-pressure cascade impactor (DLPI+, Dekati Ltd., Finland), nascent SSA particles were collected with 14 different particle size classifications (Table S2) and distributed into submicron SSA (0.016-0.94  $\mu$ m) and supermicron SSA (1.62-10  $\mu$ m) samples. Since the current study focuses on the sea-to-air transfer of DOC, all SSA samples were extracted with ultrapure water (>18.2 M $\Omega$ ·cm, 25 °C, Millipore) and the extractions were filtered with 0.45  $\mu$ m filters. Further collection details are provided in the Supplement. Blanks were prepared by unexposed quartz fiber filters with the same treatment as for SSA samples.

(3) Sampling times for seawater and the sea surface microlayer

**Page 4, lines104-105**

Seawater and sea surface microlayer (SML) samples were collected within the half-hour preceding the start of the nascent SSA generation.

(4) Determination of chlorophyll a and particulate organic carbon concentrations

**Page 4, lines 122-123**

Quantitative measurements of POC and Chl-a concentrations in seawater were carried out using the GF/F filters described in Section 2.2. Specifically, the concentration of POC in seawater was determined using an elemental analyzer (Elementar, UNICUBE), which measured the POC content in a 1 cm diameter circular area on the GF/F filter.

(5) Determination of sodium ion concentration

**Page 5, lines 131-134**

The seawater and SML samples were diluted 5,000-fold, while the submicron and supermicron SSA extracts were diluted 5-fold, ensuring that their Na+ concentrations fall within the 0.1 to 10 µg mL-1 range of a seven-point calibration curve for quantification. Repeated measurements confirmed that the relative standard deviation of the Na+ peak area remained within 6.2%.

(6) Determination of sea surface tension

**Page 5, lines 134-135**

The surface tension of filtered seawater and SML samples was measured by the platinum plate method using a surface tension meter (Powereach, JB99B, China). Each measurement was repeated three times, and the average value was taken.

**(7) Determination of fluorescent organic compounds**

The EEM results for all samples were normalized to Raman units (R. U.) by the Raman peak of water (Ex=350 nm) after subtracting the background signal obtained from Milli-Q water (Chen et al., 2023). EEM data analysis using parallel factor analysis (PARAFAC) with non-negativity constraints were performed with the DOMFlour toolbox by MATLAB R2020a (Stedmon and Bro, 2008). It is important to consider the matrix effects resulting from differences in pH and salinity between seawater samples (seawater and sea surface microlayer) and SSA samples (submicron and supermicron SSA extracts), as well as potential deviations from the variability assumptions of the PARAFAC model due to variations in DOC concentrations across the samples. Therefore, we followed the method outlined by Murphy et al. to normalize each sample's EEMs based on their total signal intensity (Murphy et al., 2013). After validating the PARAFAC model through split-half verification and random initialization analysis, the normalization was cancelled by multiplying the fractions by each sample's total signal intensity.

(8) Estimation of the uncertainty in saccharides determination

**Page 6, lines 169-171**

According a previous assessment, the desalting dialysis step retains over 90% of high-molecular-weight DOC (Engel and Händel, 2011); after acidification and hydrolysis, the average recovery rate for most saccharides ranges from 81% to 107%.

(9) Processing and gradient elution methods for DOC in high-resolution mass spectrometry analysis.

**Page 6, lines 178-181**

Water with 0.1% (v/v) formic acid (eluent A) and acetonitrile (eluent B) was applied for the SSA, SML, and seawater extractions, with a flow rate of 0.3 mL min-1. Gradient elution was performed as follows: eluent B, initially set to 5% for 4 min,

increased to 100% in 36 min, was held for 3 min, decreased to 5% in 0.5 min and was held for 12.5 min to recondition the column.

**Section 2.3.2**

Did you perform visual observations of the communities over the course of the experiment?

**Author Reply**

Although we recognize that this operation would be beneficial for deepening our understanding of the relationship between biological activity and the sea-to-air transfer, we did not perform it here. However, it will be incorporated into the experimental design in our future study.

**Section 2.3.4 Why saccharides?**

This family of compounds was not introduced in the introduction, the reason for their analysis should appear somewhere, it could be in the discussion part, but it is not clear anywhere why this specific analysis was performed.

**Author Reply**

We have made the following revisions in the revised manuscript:

In the final paragraph of the introduction, we restructured the content to seamlessly transition into the relevant sections on saccharides within the experimental methods and results and discussion sections.

**Page 3, lines 64-69**

This study explored the implications of sea-to-air transfer of DOC during phytoplankton blooms. Firstly, the macroscopic effects of phytoplankton blooms on SSA formation and DOC enrichment were examined. Secondly, we employed high-resolution mass spectrometry to analyze the molecular profiles of DOC at different stages of sea-to-air transfer. Finally, by focusing on the most significant contributors, such as proteins, saccharides, and humic substances, the patterns of DOC sea-to-air transfer during phytoplankton blooms were investigated through a micro-to-macro approach.

In the conclusion, we emphasized that studying the entry of sugars and proteins into the atmosphere via marine aerosols will have profound implications for climate.

**Page 18, lines 454-457**

Polysaccharides and amino acids produced by phytoplankton have been demonstrated to be key substances for efficient ice nucleation activity and are frequently detected in SSA and low-level clouds (Triesch et al., 2021a; Triesch et al., 2021b; Hartmann et al., 2025). Therefore, given the frequent occurrence of phytoplankton blooms and the enhancing effect of ocean warming, they will ultimately exert a profound influence on climate through the sea spray process.

The word sample is used every single sentence without further details. More precision is needed each time, what samples are mentioned? This is the same in all sections (2.3.5...)

**Author Reply**

We have revised the description of the sample in the manuscript.

**Page 6, lines 160-162**

Except for the samples collected on Days 1, 9, and 18, samples of submicron SSA, supermicron SSA, SML, and seawater collected on other days were subjected to dialysis for desalting, followed by acid hydrolysis, nitrogen blowing, and resolubilization (Engel and Händel, 2011).

**Page 6, lines 173-175**

Based on Chl-a concentration during the phytoplankton bloom, samples of submicron SSA, supermicron SSA, SML, and seawater collected on Days 1, 9 (peak of Chl-a), and 18 were pretreated for desalting and concentrating using a PPL solid-phase extraction column (100 mg/3 mL, Agilent Technologies).

**Section 2.3.5**

How can you know about the phases of the bloom without visual observations?? Clearly you could have two different successive blooms with different communities, and this is well supported by the color of your carboys. Chl-a is a weak proxy here.

**Author Reply**

We acknowledge that the stage of phytoplankton blooms cannot be determined without visual observation. Therefore, we have revised the manuscript to clarify that samples for mass spectrometry analysis were selected based solely on Chl-a concentrations.

**Page 6, lines 173-175**

Based on Chl-a concentration during the phytoplankton bloom, samples of submicron SSA, supermicron SSA, SML, and seawater collected on Days 1, 9 (peak of Chl-a), and 18 were pretreated for desalting and concentrating using a PPL solid-phase extraction column (100 mg/3 mL, Agilent Technologies).

The gradient elution should be written in the text, no table is needed in the Supplementary

**Author Reply**

We have described the elution method in Section 2.3.5 and deleted the table in the Supplement.

**Page 6, lines 178-181**

Water with 0.1% (v/v) formic acid (eluent A) and acetonitrile (eluent B) was applied for the SSA, SML, and seawater extractions, with a flow rate of 0.3 mL min-1. Gradient elution was performed as follows: eluent B, initially set to 5% for 4 min, increased to 100% in 36 min, was held for 3 min, decreased to 5% in 0.5 min and was held for 12.5 min to recondition the column (Wan et al., 2022).

[revised manuscript text omitted]

---

## Author Comment (AC3)

**Reply to the comments from Anonymous Referee #2**

We deeply appreciate Anonymous Referee #2 for the thorough review of our manuscript. Our manuscript has been revised according to the comments and our responses to the comments are as follows. For clarity, the comments are reproduced in blue, authors' responses are in black and changes in the manuscript are in red color text.

Sea spray aerosol (SSA) formation is an important pathway for the transfer of marine substances to the atmosphere. This study investigates how phytoplankton blooms promote the sea-to-air transfer of dissolved organic carbon (DOC) through SSA formation. Natural seawater was incubated outdoors to induce phytoplankton blooms, and a laboratory waterfall-type SSA simulation tank was used to reproduce the sea—air exchange process. The DOC enrichment in SSA can increase by 10–30 times during phytoplankton blooms, mainly driven by protein-like components (PRLIS), with a secondary contribution from polysaccharides modified by heterotrophic bacteria. The study is well designed and methodologically sound, covering the continuous chain from seawater to the sea surface microlayer and SSA, and it provides scientifically meaningful insights. The following suggestions are provided to the authors for further revision before the final publication.

The continuous plunging waterfall mode was adopted to improve SSA sampling efficiency; however, this configuration may not accurately reproduce the bubble dynamics and turbulence of real oceanic wave-breaking. Please discuss the representativeness and possible implications of this setup.

**Author Reply**

Plunging waterfall type has been proven as an efficient laboratory simulation method to generate sea spray aerosols (Stokes et al., 2013), and has been used widely in many research studies (Callaghan et al., 2014; Van Acker et al., 2021; Harb and Foroutan, 2022; Jayarathne et al., 2022). Collins et al. conducted a detailed comparison between intermittent and continuous plunging waterfall systems, focusing on differences in sea spray aerosol particle size distribution and organic enrichment (Collins et al., 2014). The main difference between intermittent and continuous modes

lies in how surface foam behavior affects the formation of sea spray aerosol. In intermittent operation, surface foam breaks and dissipates during operational gaps, whereas in continuous operation, foam gradually dissipates as it moves away from the impact point. At lower dissolved organic carbon concentrations ( $\approx$ 85  $\mu$ M), SSA produced by both methods shows minimal differences, while the differences become more important at higher DOC concentrations ( $\approx$ 400  $\mu$ M). However, compared to other laboratory SSA production methods (wave breaking and sintered filter glass), the differences between intermittent and continuous plunging waterfalls are relatively minor (Collins et al., 2014). During the phytoplankton bloom period, the use of a continuous plunging waterfall would help reduce our sampling time.

In the revised manuscript, we briefly discuss this configuration.

**Page 3, lines 90-94**

These two types of plunging waterfalls differ mainly in the behavior of surface bubbles as they rupture and dissipate: in intermittent waterfalls, surface bubbles breaks and dissipates during operational gaps, whereas in continuous waterfalls, surface bubbles gradually dissipates as it moves away from the impact point (Collins et al., 2014).

The enrichment factor (EF) is normalized to Na+, assuming its concentration remains constant. However, Na+ levels may vary with particle size and humidity. The authors should clarify the measurement range, precision, and variability of Na+ and discuss how potential deviations from this assumption could affect the calculated EF values.

**Author Reply**

The size of SSA is related to relative humidity (RH) as it equilibrates either by absorbing or evaporating water. At different RH, the empirical relationship among the sizes of SSA is: Dp (at 100%RH)  $\approx 2$ Dp (at 80% RH)  $\approx 4$ Dp (below 50% RH) (Veron, 2015). In other words, when RH falls below 50%, it is generally considered that water within the SSA has evaporated completely, and the diameter is no longer affected by the water content. In order to exclude the influence of RH on SSA's size, we employed a drying tube to dehumidify the airflow carrying the SSA. After drying, the airflow's

RH was kept below 30%, ensuring that the nascent SSA underwent a phase transition from liquid to solid.

Dried SSA particles were collected by a low-pressure cascade impactor (DLPI+, Dekati Ltd., Finland) and classified into submicron and supermicron SSA. Submicron SSA and supermicron SSA were extracted using 10 mL Milli-Q water. In order to quantify the Na+ concentrations, a seven-point calibration curve of 0.1 to 10 μg mL-1 range was created. Samples of seawater and sea surface microlayer were diluted 5,000-fold, and extracts of submicron SSA and supermicron SSA were diluted 5-fold, making the Na+ concentrations fall within the range of the calibration curve. The relative standard deviation of Na+ concentration after repeated measurements was controlled within 6.2%.

We have added the necessary information in the revised manuscript.

**Page 3, lines 94-98**

Nascent SSA was transported with purified air (Zero Air Supply, Model 111, Thermo Scientific), and the airflow was dried to a relative humidity below 30% (Monotube Dryer, MD700-12F-3, Perma Pure, USA) before collection and measurement. At this relative humidity, nascent SSA can be completely dried into solid particles.

Additionally, the information regarding the sample dilution, the measurement range of Na+ concentration, and the relative standard deviation ranges for multiple measurements has been updated.

**Page 5, lines 131-134**

Sodium ions (Na+) concentrations were measured using an ion chromatograph (Dionex ICS-600, Thermo Fisher Scientific, USA). The seawater and SML samples were diluted 5,000-fold, while the submicron and supermicron SSA extracts were diluted 5-fold, ensuring that their Na+ concentrations fall within the 0.1 to 10 μg mL-1 range of a seven-point calibration curve for quantification. Repeated measurements confirmed that the relative standard deviation of the Na+ peak area remained within 6.2%.

In the original manuscript, the enrichment factors for organic matter were calculated using the average of multiple measurements. In this revision, we apply uncertainty propagation formulas to incorporate measurement uncertainties into the enrichment factor results. The formulas for calculating the DOC enrichment factor and relative standard deviation (RSD) are as follows:

$$EF = \frac{\left(\mathrm{DOC}\right)_{SSA \text{ or } SML}/\left(\mathrm{Na}^{+}\right)_{SSA \text{ or } SML}}{\left(\mathrm{DOC}\right)_{SW}/\left(\mathrm{Na}^{+}\right)_{SW}}$$

$$RSD_{enrichment \text{ factor}} = \sqrt{\left(RSD_{DOC|SSA \text{ or } SML}\right)^{2} + \left(RSD_{Na}^{+}|SSA \text{ or } SML}\right)^{2} + \left(RSD_{DOC|SW}\right)^{2} + \left(RSD_{Na}^{+}|SW\right)^{2}}$$

In the revised manuscript, we have added explanations regarding the propagation of measurement uncertainty to the enrichment factor calculations and presented the specific settlement results in Figure 3.

**Page 5, lines 139-140**

Using the uncertainty transfer formula to propagate the uncertainties from multiple measurements results into the calculation of the enrichment factor.

**Figure 3.** Time series of DOC enrichment during the phytoplankton bloom. Enrichment factors of DOC relative to Na+ in the SML (purple), submicron SSA (orange) and supermicron SSA (green). The gray background is the concentration of DOC in the SML. Error bars represent the standard deviation of the EF, derived from the standard deviation of Na+ concentration and DOC concentration obtained through multiple measurements.

The fluorescence intensities used in the EEM-PARAFAC analysis may be influenced by matrix effects such as salinity, pH, and the polarity of DOM. These factors can alter fluorescence yield and spectral properties, potentially leading to biases when comparing different sample types (e.g., seawater or SSA extracts). Please include some discussions on how these matrix effects were considered or minimized, and whether they may influence the comparability of EEM results.

**Author Reply**

Regarding the potential influence of matrix effects—such as salinity, pH, and the polarity of DOM—on fluorescence intensity, these factors could indeed introduce bias between different samples (e.g., seawater versus SSA extracts), which may in turn affect the results of the EEM-PARAFAC analysis.

Changes in pH can alter the ionization state of certain fluorescent compounds, thereby affecting their fluorescence yield (Timko et al., 2015). Seawater and seawater surface microlayer samples used for EEM detection should have a pH of approximately 8.2. In contrast, extracts of submicron and supermicron SSA (extracted with 10 mL of Milli-Q water) contained very low amounts of SSA, leading to pH values close to 7 for these samples.

Salinity variations can influence the aggregation state of fluorescent molecules, thereby altering their fluorescence emission characteristics (Kholodov et al., 2024). While some studies have reported significant enrichment of divalent cations in sea surface microlayer compared to seawater (Salter et al., 2016; Schill et al., 2018), it is important to note that sodium chloride is the main contributor to seawater salinity, with divalent cations contributing relatively little. Therefore, it can be assumed that the salinity of seawater samples and that of SML samples used for EEM fluorescence detection should be quite similar. In contrast, the salinities of submicron and supermicron SSA samples' extracts are both close to and significantly lower than that of seawater.

The matrix effect can potentially influence the absolute intensity of EEM across different samples, particularly due to salinity differences between seawater-type samples (seawater and seawater micro-surface layer) and SSA-type samples

(submicron and supermicron SSA). As a result, the comparability of EEM's absolute intensities between these two categories requires careful verification. Therefore, in the manuscript, our discussion of EEM's absolute intensity in Figure 5(b) is limited to comparisons within each category—either between seawater and sea surface microlayer, or between submicron and supermicron SSA.

When performing EEM-PARAFAC analysis, we accounted for matrix effects and variations in DOC concentrations between seawater and SSA samples. Following the methodology outlined by Murphy et al. (Murphy et al., 2013), we normalized the EEM for each sample to its total signal using the signal normalization module within the drEEM toolbox. This normalization ensures that each sample's EEM contributes equally during the PARAFAC analysis, enabling the model to focus on chemical variations between samples rather than the overall signal magnitude. Additionally, this approach enhances the detection of minor peaks. After model validation, normalization can be cancelled by multiplying the fractions by each sample's total signal intensity.

A discussion on matrix effects, emphasizing their potential impact on fluorescence spectra and acknowledging that these variability sources were considered during data analysis was updated in the revised manuscript

**Page 5, lines 144-154**

The excitation-emission matrix (EEM) of DOC was obtained using a fluorescence and absorbance spectrometer (DuettaTM, Horiba Scientific, Japan). The excitation wavelength of EEM was in the range of 250-620 nm, the emission wavelength was in the range of 250-700 nm, the scanning intervals were set to 5 nm and 2 nm, respectively, and the slit width was fixed at 5 nm. The EEM results for all samples were normalized to Raman units (R. U.) by the Raman peak of water (Ex=350 nm) after subtracting the background signal obtained from Milli-Q water (Chen et al., 2023). EEM data analysis using parallel factor analysis (PARAFAC) with non-negativity constraints were performed with the DOMFlour toolbox by MATLAB R2020a (Stedmon and Bro, 2008). It is important to consider the matrix effects resulting from differences in pH and salinity between seawater samples (seawater and sea surface microlayer) and SSA samples (submicron and supermicron SSA extracts), as well as potential deviations

from the variability assumptions of the PARAFAC model due to variations in DOC concentrations across the samples. Therefore, we followed the method outlined by Murphy et al. to normalize each sample's EEMs based on their total signal intensity (Murphy et al., 2013). After validating the PARAFAC model through split-half verification and random initialization analysis, the normalization was cancelled by multiplying the fractions by each sample's total signal intensity.

The method recovery, reproducibility, and the detection limit of organic species are suggested to be provided in the method.

**Author Reply**

We have added this information to the Methods section of the manuscript:

(1) We provided additional details on the dilution factor for the samples, the concentration range of the standard curve, and the relative standard deviation of repeated measurements when measuring Na+ concentration using ion chromatography.

**Page 5, lines 131-134**

Sodium ions (Na+) concentrations were measured using an ion chromatograph (Dionex ICS-600, Thermo Fisher Scientific, USA). The seawater and SML samples were diluted 5,000-fold, while the submicron and supermicron SSA extracts were diluted 5-fold, ensuring that their Na+ concentrations fall within the 0.1 to 10 µg mL-1 range of a seven-point calibration curve for quantification. Repeated measurements confirmed that the relative standard deviation of the Na+ peak area remained within 5.2%.

(2) The number of repetitions for the surface tension measurements, previously described in the caption of Figure 2, has been added to the Methods section.

**Page 5, lines 135-136**

The surface tension of filtered seawater and SML samples was measured by the platinum plate method using a surface tension meter (Powereach, JB99B, China). Each measurement was repeated three times, and the average value was taken.

(3) In the EEM-PARAFAC method, we have clarified how matrix effects are accounted for during data processing and how variations in dissolved organic carbon concentrations across different samples are addressed.

**Page 5, lines 146-154**

EEM data analysis using parallel factor analysis (PARAFAC) with non-negativity constraints were performed with the DOMFlour toolbox by MATLAB R2020a (Stedmon and Bro, 2008). It is important to consider the matrix effects resulting from differences in pH and salinity between seawater samples (seawater and sea surface microlayer) and SSA samples (submicron and supermicron SSA extracts), as well as potential deviations from the variability assumptions of the PARAFAC model due to variations in DOC concentrations across the samples. Therefore, we followed the method outlined by Murphy et al. (Murphy et al., 2013) to normalize each sample's EEMs based on their total signal intensity. After validating the PARAFAC model through split-half verification and random initialization analysis, the normalization was cancelled by multiplying the fractions by each sample's total signal intensity.

(4) In Section 2.3.5, we supplemented the average recovery rate for carbohydrate detection.

**Page 6, lines 168-170**

The quantification was performed using seven-point standardized calibration curves with concentrations ranging from 10 nM to 10  $\mu$ M. According a previous assessment, the desalting dialysis step retains over 90% of high-molecular-weight DOC (Engel and Händel, 2011); after acidification and hydrolysis, the average recovery rate for most saccharides ranges from 81% to 107%.

Please specify in the abstract whether the reported "10-fold to 30-fold enrichment" of DOC refers to SW or to the SML.

**Author Reply**

We have included the specification in the revised abstract.

**Page 1, lines 17-19**

In this study, we observed that the phytoplankton bloom can promote DOC

enrichment in SSA by 10- to 30-fold compared to seawater and investigated the mechanism of DOC sea-to-air transfer using various characterization tools.

Ensure consistent color scales in Figure 5 EEM panels to enable visual comparison.

**Author Reply**

After careful consideration of the Referee's suggestion and reviewing relevant literature, we have decided to retain the different color scales. The specific reasons are as follows:

Figure 5a shows the three types of fluorescent chromophores, which coexist in seawater, the SML, submicron SSA, and supermicron SSA, identified through the EEM-PARAFAC method. Since the identification of fluorescent chromophores relies primarily on the excitation and emission wavelengths corresponding to the peak fluorescence signals, selecting an appropriate color scale is crucial for accurately depicting the positions of these peaks.

The fluctuations in both absolute and relative fluorescence intensities of these three fluorescent substances during different phytoplankton blooms are shown in Figures 5b and 5c, allowing for visual comparison.

**Figure 5.** Sea-to-air transfer of HULIS and PRLIS. Three organics identified using the EEM-PARAFAC method: (a) HULIS 1, HULIS 2, and PRLIS. (b) EEM intensities of the three organics in different samples with respect to time. Note that in order to exclude the effect of SSA collection mass on EEM intensity, EEM intensities of SSA samples were normalized with their Na+ concentrations. (c) Relative abundance of EEM intensities of the three organics in different samples with respect to time. (d) Spearman's correlation between Chl-a, DOC and POC concentrations in seawater, POC concentration in the SML and EEM intensities of three fluorescent substances.

Add standard deviation or error bars for EF values to better exhibit measurement uncertainty.

**Author Reply**

Measurement uncertainties into the enrichment factor results were obtained by applying the uncertainty propagation formulas. These formulas for calculating the DOC enrichment factor and relative standard deviation (RSD) are as follows:

$$EF = \frac{\left(DOC\right)_{SSA \text{ or } SML}/\left(Na^{^{+}}\right)_{SSA \text{ or } SML}}{\left(DOC\right)_{SW}/\left(Na^{^{+}}\right)_{SW}}$$

$$RSD_{enrichment \text{ factor}} = \sqrt{\left(RSD_{DOC|SSA \text{ or } SML}\right)^{2} + \left(RSD_{Na^{^{+}}|SSA \text{ or } SML}\right)^{2} + \left(RSD_{DOC|SW}\right)^{2} + \left(RSD_{Na^{^{+}}|SW}\right)^{2}}$$
 We have added error bars to Figure 3 and updated the figure.

**Figure 3.** Time series of DOC enrichment during the phytoplankton bloom. Enrichment factors of DOC relative to Na+ in the SML (purple), submicron SSA (orange) and supermicron SSA (green). The gray background is the concentration of DOC in the SML. Error bars represent the standard deviation of the EF, derived from the standard deviation of Na+ concentration and DOC concentration obtained through multiple measurements.

**References**

Callaghan, A. H., Stokes, M. D., and Deane, G. B.: The effect of water temperature on air entrainment, bubble plumes, and surface foam in a laboratory breaking-wave analog, J. Geophys. Res. Oceans, 119, 7463-7482, 2014.

Chen, H., Yan, C., Fu, Q., Wang, X., Tang, J., Jiang, B., Sun, H., Luan, T., Yang, Q., Zhao, Q., Li, J., Zhang, G., Zheng, M., Zhou, X., Chen, B., Du, L., Zhou, R., Zhou, T., and Xue, L.: Optical properties and molecular composition of wintertime atmospheric water-soluble organic carbon in different coastal cities of eastern China, Sci. Total Environ., 892, 164702,

- https://doi.org/10.1016/j.scitotenv.2023.164702, 2023.
- Collins, D. B., Zhao, D. F., Ruppel, M. J., Laskina, O., Grandquist, J. R., Modini, R. L., Stokes, M. D., Russell, L. M., Bertram, T. H., Grassian, V. H., Deane, G. B., and Prather, K. A.: Direct aerosol chemical composition measurements to evaluate the physicochemical differences between controlled sea spray aerosol generation schemes, Atmos. Meas. Tech., 7, 3667-3683, 2014.
- Engel, A. and Händel, N.: A novel protocol for determining the concentration and composition of sugars in particulate and in high molecular weight dissolved organic matter (HMW-DOM) in seawater, Mar. Chem., 127, 180-191, <a href="https://doi.org/10.1016/j.marchem.2011.09.004">https://doi.org/10.1016/j.marchem.2011.09.004</a>, 2011.
- Harb, C. and Foroutan, H.: Experimental development of a lake spray source function and its model implementation for Great Lakes surface emissions, Atmos. Chem. Phys., 22, 11759-11779, https://doi.org/10.5194/acp-22-11759-2022, 2022.
- Jayarathne, T., Gamage, D. K., Prather, K. A., and Stone, E. A.: Enrichment of saccharides at the airwater interface: a quantitative comparison of sea surface microlayer and foam, Environ. Chem., 19, 506-516, <a href="https://doi.org/10.1071/EN22094">https://doi.org/10.1071/EN22094</a>, 2022.
- Kholodov, V. A., Danchenko, N. N., Ziganshina, A. R., Yaroslavtseva, N. V., and Semiletov, I. P.: Direct Salinity Effect on Absorbance and Fluorescence of Chernozem Water-Extractable Organic Matter, Aquatic Geochemistry, 30, 31-48, 10.1007/s10498-024-09423-w, 2024.
- Murphy, K., Stedmon, C., Graeber, D., and Bro, R.: Fluorescence spectroscopy and multi-way techniques. PARAFAC, Anal. Methods, 5, 6541–6882, 10.1039/c3ay41160e, 2013.
- Salter, M. E., Hamacher-Barth, E., Leck, C., Werner, J., Johnson, C. M., Riipinen, I., Nilsson, E. D., and Zieger, P.: Calcium enrichment in sea spray aerosol particles, Geophys. Res. Lett., 43, 8277-8285, 2016.
- Schill, S. R., Burrows, S. M., Hasenecz, E. S., Stone, E. A., and Bertram, T. H.: The Impact of Divalent Cations on the Enrichment of Soluble Saccharides in Primary Sea Spray Aerosol, Atmosphere, 9, 476, 2018.
- Stedmon, C. A. and Bro, R.: Characterizing dissolved organic matter fluorescence with parallel factor analysis: a tutorial, Limnol. Oceanogr. Meth., 6, 572-579, <a href="https://doi.org/10.4319/lom.2008.6.572">https://doi.org/10.4319/lom.2008.6.572</a>, 2008.
- Stokes, M. D., Deane, G. B., Prather, K., Bertram, T. H., Ruppel, M. J., Ryder, O. S., Brady, J. M., and Zhao, D.: A Marine Aerosol Reference Tank system as a breaking wave analogue for the production of foam and sea-spray aerosols, Atmos. Meas. Tech., 6, 1085-1094, https://doi.org/10.5194/amt-7-3667-2014, 2013.
- Timko, S. A., Gonsior, M., and Cooper, W. J.: Influence of pH on fluorescent dissolved organic matter photo-degradation, Water Res., 85, 266-274, <a href="https://doi.org/10.1016/j.watres.2015.08.047">https://doi.org/10.1016/j.watres.2015.08.047</a>, 2015.
- Van Acker, E., Huysman, S., De Rijcke, M., Asselman, J., De Schamphelaere, K. A. C., Vanhaecke, L., and Janssen, C. R.: Phycotoxin-Enriched Sea Spray Aerosols: Methods, Mechanisms, and Human Exposure, Environ. Sci. Technol., 55, 6184-6196, 10.1021/acs.est.1c00995, 2021.
- Veron, F.: Ocean Spray, Annu. Rev. Fluid Mech., 47, 507-538, <a href="https://doi.org/10.1146/annurev-fluid-010814-014651">https://doi.org/10.1146/annurev-fluid-010814-014651</a>, 2015.